# Topology-Aware Conformal Prediction for Stream Networks

**Jifan Zhang** [*]
Northwestern University
Evanston, IL 60208
jifanzhang2026@u.northwestern.edu

**Fangxin Wang** [*]
University of Illinois Chicago
Chicago, IL 60607
fwang51@uic.edu

**Zihe Song**
University of Illinois Chicago
Chicago, IL 60607
zsong29@uic.edu

**Philip Yu**
University of Illinois Chicago
Chicago, IL 60607
psyu@uic.edu

**Kaize Ding** [†]
Northwestern University
Evanston, IL 60208
kaize.ding@northwestern.edu

**Shixiang Zhu** [†]
Carnegie Mellon University
Pittsburgh, PA 15213
shixiangzhu@cmu.edu

## Abstract

Stream networks, a unique class of spatiotemporal graphs, exhibit complex directional flow constraints and evolving dependencies, making uncertainty quantification a critical yet challenging task. Traditional conformal prediction methods struggle in this setting due to the need for joint predictions across multiple interdependent locations and the intricate spatio-temporal dependencies inherent in stream networks. Existing approaches either neglect dependencies, leading to overly conservative predictions, or rely solely on data-driven estimations, failing to capture the rich topological structure of the network. To address these challenges, we propose Spatio-Temporal Adaptive Conformal Inference (STACI), a novel framework that integrates network topology and temporal dynamics into the conformal prediction framework. STACI introduces a topology-aware nonconformity score that respects directional flow constraints and dynamically adjusts prediction sets to account for temporal distributional shifts. We provide theoretical guarantees on the validity of our approach and demonstrate its superior performance on both synthetic and real-world datasets. Our results show that STACI effectively balances prediction efficiency and coverage, outperforming existing conformal prediction methods for stream networks.

## 1 Introduction

Stream networks represent a distinctive class of spatiotemporal graphs where data observations follow directional pathways and evolve dynamically over both space and time [21]. These networks are prevalent in various domains such as hydrology, transportation, and environmental monitoring, where data exhibit strong flow constraints [9, 18, 33, 57, 36]. For example, in hydrology, river networks dictate the movement of water flow and pollutant dispersion [32], while in transportation, road and rail networks determine congestion and travel times [57, 56]. Understanding and modeling these networks are crucial for infrastructure planning, disaster response, and ecological conservation.

---

[*]Equal Contribution.
[†]Co-corresponding Authors.

39th Conference on Neural Information Processing Systems (NeurIPS 2025).

A fundamental challenge in stream network analysis is predicting future observations and quantifying their uncertainty across multiple interconnected locations governed by network topology. Given the dynamic nature of these systems, accurate and reliable uncertainty quantification (UQ) is essential for risk assessment, decision-making, and resource allocation. For example, in transportation, estimating uncertainty in traffic volume forecasts across critical junctions enables optimal routing and congestion management [57]. However, the hierarchical dependencies, directional flow constraints, and evolving conditions inherent in stream networks introduce significant complexities.

Recent advances in machine learning and statistical modeling have enhanced predictive accuracy for spatio-temporal data and enabled effective UQ with statistical guarantees [49, 14, 55]. In particular, conformal prediction (CP) has emerged as a powerful UQ framework, providing finite-sample validity guarantees under mild assumptions [35]. By constructing prediction sets with valid coverage probabilities, CP ensures that future observations fall within specified confidence intervals, enhancing reliability in decision-support systems [27, 23, 5, 54].

Despite its success in various domains, traditional CP methods face significant limitations when applied to stream networks due to two key challenges: (*i*) *Multivariate prediction*: Unlike standard time-series predictions that focus on a single target variable, stream networks require joint predictions at multiple locations, where observations are highly interdependent. Applying CP independently at each location neglects network-wide dependencies, leading to inefficiencies in prediction set construction and potential loss of coverage guarantees. (*ii*) *Intricate spatio-temporal flow constraints*: Traditional CP assumes exchangeable data, an assumption that fails in stream networks due to directional flow constraints. While graph-based and spatial models account for topological relationships, stream networks exhibit unique dependency structures that neither conventional graph-based approaches nor purely data-driven models fully capture. Existing CP approaches either completely ignore dependencies without considering the spatio-temporal dynamics [37, 10] or attempt to learn dependencies solely from data without incorporating topological constraints [51, 40]. The former results in overly conservative or miscalibrated prediction sets, while the latter risks overfitting to specific network conditions, reducing generalizability.

To address these challenges, we propose a novel framework, Spatio-Temporal Adaptive Conformal Inference (`STACI`), for constructing uncertainty sets in stream networks. Our method integrates network topology and temporal dynamics into the conformal prediction framework, yielding more efficient and reliable UQ. Specifically, we develop a nonconformity score that explicitly incorporates spatial dependencies across multiple locations on the stream network as determined by their underlying topology, balancing observational correlations with topology-induced dependencies. To achieve this balance, we introduce a weighting parameter that regulates the contribution of topology-based covariance and data-driven estimates. A greater reliance on the topology-induced covariance structure improves coverage guarantees, assuming it accurately reflects underlying dependencies. Conversely, prioritizing sample-based estimates mitigates potential misspecifications in the topology-induced covariance, often leading to better predictive efficiency. Additionally, we consider a dynamic adjustment mechanism that accounts for temporal distributional shifts, allowing prediction intervals to adapt over time and maintain valid coverage in non-stationary environments.

We provide a theoretical analysis of `STACI`, demonstrating that it maximizes prediction efficiency by reducing uncertainty set volume while maintaining valid coverage guarantees. To validate its effectiveness, we evaluate `STACI` on synthetic data with a stationary covariance matrix and real-world data with time-varying covariance, comparing its performance against state-of-the-art baseline. [3] Both our theoretical and empirical results underscore the importance of the weighting parameter that balances data-driven insights with topology-induced knowledge, optimizing performance and enhancing predictive reliability in stream network applications.

Our contribution can be summarized as follows:

- We propose a novel conformal prediction framework specifically designed for stream networks, integrating both spatial topology and temporal dynamics to enhance uncertainty quantification.
- We highlight the limitations of purely data-driven dependency estimation in stream networks and introduce a principled approach that leverages both observational data and inherent network structure.

---

[3]Our code is publicly available at `https://github.com/fangxin-wang/STCP`.

- We provide a theoretical analysis establishing STACI's validity and efficiency, and empirically demonstrate its superior performance in achieving an optimal balance between coverage and prediction efficiency on both synthetic and real-world datasets.

## 2 Related Works

Stream networks, such as hydrology [21, 18, 36], transportation networks [14, 28], and environmental science networks [25], have been extensively studied due to their critical role in natural and engineered systems. Forecasting for stream network can be approached from two perspectives: as a graph prediction problem or as a multivariate time series prediction problem. In this work, we focus on the latter one, with the aim of predicting future data based on historical network data.

Many approaches to stream network analysis rely on domain-specific statistical and physical models. [18] introduced spatial stream network models for hydrology, emphasizing the importance of flow-connected relationships and spatial autocorrelation. The tail-up model [43] generalized spatial covariance structures to stream networks by weighting observations based on flow connectivity. Recent advances in machine learning have spurred innovative approaches for modeling stream networks for point forecasts, particularly through graph-based frameworks that leverage their inherent spatio-temporal (ST) graph dynamics [20, 11].

While effective and widely adopted, models without uncertainty quantification often lack considerations for reliability, posing limitations particularly in safety-critical scenarios. To address this, some studies [49, 58, 38, 48, 33] turn to explore interval prediction, which ensures that prediction intervals cover the ground-truth values with a pre-defined high probability, offering a more reliable alternative. Among these approaches, the majority of studies employ Bayesian methods to construct prediction intervals for ST forecasting problems [47]. These methods commonly utilize Monte Carlo Dropout [49, 33] or Probabilistic Graph Neural Networks [58, 48]. However, the performance of Bayesian methods has been found to be sensitive to the choice of prediction models and priors, particularly the type of probabilistic distributions [48]. To address these limitations, classic Frequentist-based methods have been employed, such as conformal prediction, which generally offer more robust coverage across data and model variations.

Conformal prediction (CP) [46] has recently gained significant attention across various domains, including graph-structured data [8, 19, 29] and multi-dimensional time series [40, 30, 50]. Existing CP methods for graphs [8, 19, 29] have primarily addressed node/edge classification and ranking tasks, where UQ concerns discrete labels or scores on static graph structures. In contrast, our work focuses on a fundamentally different problem: UQ for spatio-temporal forecasting on dynamic networks. Given that spatio-temporal graphs can be naturally formulated as a special case of multivariate time series, we discuss related CP methodologies developed for the latter setting [40, 30, 50]. [40] assumes that data samples for each entire time series are drawn independently from the same distribution, while [30] assumes exchangeability in the data. Both approaches fail to capture the complex temporal and spatial dependencies inherent in ST graphs, limiting their applicability. [50] construct ellipsoidal prediction regions for non-exchangeable multi-dimensional time series, but their model neglects the inherent graph structure embedded within the multi-dimensional time series and overlook scenarios where the error process (see eq. (1)) is non-stationary, a prevalent feature in real-world data. Section B in the Appendix provides a taxonomy of existing CP methods, highlighting our unique positioning within the CP literature. To the best of our knowledge, no previous work has specifically tailored CP for stream networks or other spatio-temporal graphs, reinforcing the novelty to this domain.

## 3 Problem

Consider a stream network $\mathcal{G}$ with fixed flow direction at time $t \in \{1, \ldots, T\}$, with $I$ observational sites indexed by $\mathcal{I} = \{1, \ldots, I\}$. Let $\mathcal{L} \subset \mathbb{R}^2$ denote the set of all geolocations on the network, and let the geographical location of site $i \in \mathcal{I}$ be represented as $\ell_i \in \mathcal{L}$. The stream network consists of segments $\{r_j \subset \mathcal{L}, j \in \mathcal{J}\}$, where $\mathcal{J}$ is the index set of all stream segments. Each site $i \in \mathcal{I}$ is located within a specific segment $r_j$ for some $j \in \mathcal{J}$, and a segment may contain multiple or no observational sites. For any location $u \in \mathcal{L}$, we define $\wedge u$ as the set of all upstream segments of location $u$, and $\vee u$ as the set of all downstream segments of location $u$. The hydrologic distance between two locations $v, u \in \mathcal{L}$, denoted as $d(v, u)$, is the distance measured along the stream. If $v$ and $u$ belong to the same segment $r_i$, $d(v, u)$ is simply the Euclidean distance between $v$ and $u$. See Figure 1 for an illustration.

Now, consider a multivariate time series observed at the $I$ sites. We denote the dataset as $\mathcal{D} := \{(X_t, Y_t)\}_{t \in [T]}$. Here $Y_t := [Y_t(\ell_1), Y_t(\ell_2), \ldots, Y_t(\ell_I)]^\top \in \mathbb{R}^I$, and $Y_t(\ell_i)$ (or simply

$Y_t^i$) represents the observation at location $\ell_i$ at time $t$. The historical observations are given by $X_t \in \mathbb{R}^{I \times h}$, defined as $X_t := [Y_{t-1}, Y_{t-2}, \ldots, Y_{t-h}]^\top \in \mathbb{R}^{I \times h}$. We assume that $Y_t$ follows an unknown true model $f(X_t)$ with additive noise $\epsilon_t$, such that:

$$Y_t = f(X_t) + \epsilon_t, \tag{1}$$

where $\epsilon_t \in \mathbb{R}^I$ has zero mean and a positive definite covariance matrix $\Sigma \succ 0$.

The goal is to construct a prediction set for $Y_{T+1}$ given the new history $X_{T+1}$, denoted by $\mathcal{C}(X_{T+1})$, such that, for a predefined confidence level $\alpha$, the following coverage guarantee holds:

$$P(Y_{T+1} \in \mathcal{C}(X_{T+1})) \geq 1 - \alpha.$$

This objective can be achieved using split conformal prediction (CP) [46], a widely used framework for uncertainty quantification. Split CP operates by first partitioning the data into a training set and a *calibration* set. The prediction model $\hat{f}$ is trained exclusively on the training set. To assess the reliability of predictions, a *nonconformity score* is computed, which quantifies the deviation of each calibration sample from the ground truth. Given a target confidence level $\alpha$, the method determines the $(1-\alpha)$-th quantile of the nonconformity scores from the calibration data. This quantile is then used to adjust $\hat{f}$'s predictions for test samples, ensuring the constructed prediction sets maintain valid

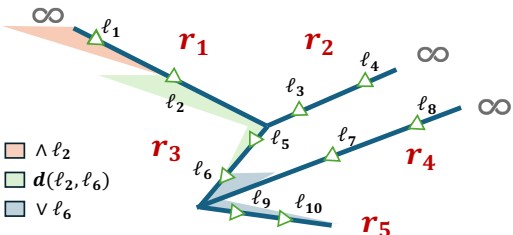

Figure 1: An example of stream network $\mathcal{G}$. The network segments $\{r_1, \ldots, r_5\}$ are denoted by blue lines, and the observation points $\{\ell_1, \ldots, \ell_{10}\}$ are marked with green triangles, pointing to the flow directions. The upstream of location $\ell_2$ are segments accompanied by orange area, and the downstream of location $\ell_6$ are blue shaded. The hydrologic distance between $\ell_2$ and $\ell_6$ is calculated through adding lengths of green shaded segments in both $r_1$ and $r_3$.

coverage. Under the assumption that the calibration and test data are exchangeable, the prediction sets are guaranteed to achieve a coverage rate of at least $1 - \alpha$ on the test data.

The challenges of performing multivariate time-series prediction over a stream network are twofold: (*i*) *Multi-dimensionality*: The response variable $Y_t$ is multivariate and potentially high-dimensional, significantly increasing the complexity of constructing accurate prediction sets. Standard CP methods, when applied to multi-dimensional variables without a carefully designed nonconformity score, often produce overly conservative prediction sets. This leads to inefficiencies, as the prediction set size $|\mathcal{C}(X_t)|$ becomes too large to provide meaningful uncertainty quantification. (*ii*) *Non-exchangeability*: Observational sites exhibit complex spatial and temporal dependencies due to strong correlations imposed by the network topology. As a result, traditional CP methods, which rely on exchangeability assumptions, cannot be readily applied.

## 4 Proposed Framework

This paper proposes a novel framework, referred to as spatio-temporal adaptive conformal inference (STACI), for constructing uncertainty sets in spatio-temporal stream networks. Our approach consists of two key components: (*i*) We develop a nonconformity score that explicitly captures spatial dependencies induced by the stream network's topology, leading to more efficient prediction sets. (*ii*) We account for temporal distributional shifts to refine prediction sets dynamically, ensuring reliable coverage over time. We demonstrate that STACI significantly improves prediction efficiency while maintaining valid coverage guarantees, making it a robust and effective approach for uncertainty quantification in spatio-temporal settings.

**Topology-aware Nonconformity Score** We use the most recent $n < T$ data points to construct the calibration dataset. Specifically, we denote the calibration dataset as $\mathcal{D}_{\text{cal}} := \{(X_t, Y_t), t = T - n + 1, \cdots T - 1, T\}$, and define $\hat{Y}_t := \hat{f}(X_t)$, where $\hat{f}$ is the fitted model trained on the rest of the data $\mathcal{D} \setminus \mathcal{D}_{\text{cal}}$. For each calibration data point $(X_t, Y_t) \in \mathcal{D}_{\text{cal}}$, we compute its nonconformity score, denoted by $s(X_t, Y_t)$.

To account for the intricate spatio-temporal dependencies, we consider a general class of nonconformity score functions based on the Mahalanobis distance [22]:

$$s(X_t, Y_t) := \hat{\epsilon}_t^\top A \hat{\epsilon}_t, \quad \forall t \in \mathcal{D}_{\text{cal}}, \tag{2}$$

where $A$ is an $I \times I$ symmetric positive definite matrix and $\hat{\epsilon}_t := Y_t - \hat{Y}_t - \bar{\epsilon}_t$ is the centered prediction error, with $\bar{\epsilon}_t$ denoting the sample average of errors on $\mathcal{D}_{\text{cal}}$.

The core idea of our method is a linearly weighted representation for $A$, which integrates both topology-induced and sample-based covariance estimates. Formally,

$$A := (1 - \lambda)\hat{\Sigma}_n^{-1} + \lambda \hat{\Sigma}_{\mathcal{G}}^{-1}. \tag{3}$$

Here $\hat{\Sigma}_n$ is the sample covariance matrix computed from the residuals $\{\hat{\epsilon}_t, t \in \mathcal{D}_{\text{cal}}\}$, and $\hat{\Sigma}_{\mathcal{G}}$ represents the covariance structure induced by the stream network topology. The weighting parameter $\lambda \in [0, 1]$ balances these two estimates. A higher value of $\lambda$ places greater reliance on the topology-driven covariance structure, assuming it accurately captures the underlying dependencies. Conversely, a lower $\lambda$ shifts reliance toward the sample-based estimate, mitigating potential misspecifications in the topology-induced covariance.

Unlike the method proposed by [51], which relies solely on the sample covariance estimate, this formulation incorporates the underlying topology of the stream network. By balancing data-driven and structural information, it provides a more robust covariance estimation, leading to better prediction efficiency without sacrificing coverage validity.

**Topology-induced Covariance Estimation**    We develop a novel method to estimate the topology-induced covariance $\hat{\Sigma}_{\mathcal{G}}$ used in eq. (3) by assuming the observations on the stream network can be captured by a tail-up model [18, 43, 17]. The tail-up model is formally defined as follows:

**Definition 1** (Tail-up model). *Given a stream network $\mathcal{G}$, the observation at any location $u$ on the network can be modeled as a white-noise random process, which is constructed by integrating a moving average function over the upstream process, i.e.,*

$$Y(u) = \mu(u) + \int_{\wedge u} m(v - u)\sqrt{\frac{w(v)}{w(u)}} \, dB(v), \tag{4}$$

*where $\wedge u$ denotes all the segments that are the upstream of $u$. Here, $\mu(u)$ is the deterministic mean at $u$, and $m(v - u)$ is a moving average function capturing the influence from upstream location $v$ to $u$. Both $w(v)$ and $w(u)$ are weights that satisfy the additivity constraint such that the variance remains constant across sites.*

We note that the tail-up model only requires the assumptions of ergodicity and spatial stationarity [42], which is highly flexible and can be broadly applied to a wide range of stream network data. Also, the choice of the moving average function $m(\Delta)$ remains adaptable, as long as it has a finite volume, allowing the model to accommodate different spatial structures effectively.

To estimate $\hat{\Sigma}_{\mathcal{G}}$, we model $B(v)$ using Brownian motion and adopt an exponential moving average function for $m(\Delta) = \beta \exp(-\Delta/\phi)$. Therefore, the topology-induced covariance between any two locations $u, v$ can be expressed as follows (See Lemma 5 in Section A of the Appendix):

$$\hat{\Sigma}_{\mathcal{G}}(u, v) = \sigma^2 \sqrt{\frac{w(u)}{w(v)}} \exp\left(-d(u, v)/\phi\right), \text{ if } u \to v, \tag{5}$$

where $\phi$ and $\sigma^2$ are estimated scaling parameters of the tail-up model. In practice, weights $w$ can be obtained by estimating the intensity of the flow through the observational, for instance, using normalized traffic counts as the weights for traffic stream network data.

Intuitively, the covariance structure reflects how information propagates along the stream network. The exponential decay in eq. (5) models diminishing influence with increasing hydrologic distance $d(u, v)$, while the weight term $\sqrt{w(u)/w(v)}$ modulates this effect based on flow intensity. This formulation naturally aligns with real-world stream dynamics, where proximal upstream sites exert stronger influence than distant or disconnected ones.

**Adaptive Uncertainty Set Construction**    We construct a spatio-temporally adaptive prediction set for a new observed history $X_{T+1}$ using our proposed nonconformity score, defined in eq. (2), as follows:

$$\mathcal{C}(X_{T+1}; \alpha) = \{y : s(X_{T+1}, y) \leq \hat{Q}_{1-\alpha}\},$$

where $\hat{Q}_{1-\alpha}$ is the $(1 - \alpha)$-th quantile of the empirical cumulative distribution function of $\{s(X_t, Y_t), t \in \mathcal{D}_{cal}\}$.

To account for potential temporal distribution shifts in the predictive error of eq. (1), we adopt the Adaptive Conformal Inference (ACI) framework proposed in [15]. This approach dynamically updates the confidence level $\alpha_t$ over time, ensuring that the prediction set remains responsive to evolving data distributions. Specifically, we iteratively update $\alpha_t$, and reconstruct the prediction set $C(X_t, \alpha_t)$ accordingly. At the initial test time $T + 1$, the confidence level is set as $\alpha_{T+1} = \alpha$. For subsequent time steps $t > T + 1$, $\alpha_t$, we update $\alpha_t$ with a step size $\gamma \geq 0$ as follows:

$$\alpha_{t+1} = \alpha_t + \gamma(\alpha - \mathbb{1}\{Y_t \notin C(X_t; \alpha_t)\}), \quad \forall t \geq T + 1. \tag{6}$$

The rationale behind ACI is that if the prediction set fails to cover the true value at time $t$, the effective error level is reduced, leading to a wider prediction interval at time $t + 1$, thereby increasing the likelihood of coverage. A larger step size $\gamma$ makes the method more responsive to observed distribution shifts but also introduces greater fluctuations in $\alpha_t$. When $\gamma = 0$, the method reduces to standard conformal prediction with fixed $\alpha$. The detailed analysis of coverage guarantee of Adaptive Uncertainty Set construction is discussed in Section C in Appendix.

## 5 Theoretical Analysis

Our theoretical analysis focuses on establishing two key properties for the proposed `STACI`:

1. Optimal Efficiency: We establish that `STACI` maximizes predictive efficiency by reducing the uncertainty set volume, justifying the need for accurate covariance estimation in spatio-temporal stream networks (Theorem 2).
2. Validity Guarantees under Stationarity and Adaptation to Distribution Shifts: We prove that `STACI` ensures valid conditional coverage under stationary assumptions (Theorem 1) and extend the framework to handle non-stationary settings via an ACI adjustment, ensuring approximate average coverage (Proposition 1 in Section C of the Appendix).

Our analysis is based on the Mahalanobis distance framework in eq. (2), which enables the construction of arbitrary ellipsoidal uncertainty sets, with greater flexibility for nonconformity scores. For example, standard CP with spherical uncertainty sets arises as a special case when $A$ is an identity matrix. Another instance is the approach in [51], where $A$ is set as the sample covariance matrix.

We adopt standard asymptotic notation and norm definitions. The big-$\mathcal{O}$ notation $\mathcal{O}(\cdot)$ characterizes an upper bound on a function's growth rate: if $f(n) = \mathcal{O}(g(n))$, then there exists a positive constant $C$ such that $f(n) \leq Cg(n)$, for all $n \geq n_0$. The little-$o$ notation $o(\cdot)$ denotes strictly smaller asymptotic growth, with $f(n) = o(g(n))$ implying $\lim_{n \to \infty} f(n)/g(n) = 0$. Additionally, we use standard $\ell_2$ norms for quantifying vector and matrix magnitudes.

### 5.1 Coverage Validity

We analyze the conditional coverage validity of the proposed method. Consider the additive error model described in eq. (1) where the errors, $\epsilon_t$, are *i.i.d.*. We introduce the following assumption and, for simplicity, denote the nonconformity score $\epsilon_t^\top A \epsilon_t$ as $s_t$.

**Assumption 1** (Estimation quality). *There exists a sequence $\{\nu_n\}$, $n \geq 1$ such that $\frac{1}{n} \sum_{t=T-n+1}^{T} \|\epsilon_t - \hat{\epsilon}_t\|^2 \leq \nu_n^2, \|\epsilon_{T+1} - \hat{\epsilon}_{T+1}\| \leq \nu_n$ .*

**Remark 1.** *The assumption ensures that the prediction error is bounded by $\nu_n^2$. For many estimators, the $\nu_n$ vanishes as $n \to \infty$, indicating improved estimation accuracy with larger sample sizes [7].*

**Assumption 2** (Convergence of $A_n$). *The sequence $\{A_n\}$ associated with the nonconformity score converges to a fixed matrix $A$ as $n$ increases, with an upper-bounded convergence rate $o(g(n))$, i.e. $\|A_n - A\| = o(g(n))$. Additionally, there exists a constant $r > 0$ such that $\|A\| \leq r$.*

**Remark 2.** *When designing nonconformity scores, the matrix $A$ can be chosen to either remain constant or converge to a fixed matrix. The flexibility in selecting $A$ allows for adaptability across different scenarios. For example, if the true covariance matrix of the error $\epsilon$ is known, $A$ can be set as its inverse. Alternatively, if only sample estimates are available, $A$ can be chosen as the inverse of the estimated sample covariance matrix of $\epsilon$, provided it converges under proper tail behavior conditions [45]. The major difference between choices of $A_n$ lies in the respective convergence rates.*

**Assumption 3** (Regularity conditions for $s_t$ and $\epsilon_t$). *Assume that the cumulative distribution function (CDF) of the true nonconformity score, $F_s(x)$, is Lipschitz continuous with a constant $L > 0$. Suppose there exist constants $\kappa_1, \kappa_2 > 0$ such that $\|\epsilon_t\| \leq \kappa_1 I$ almost surely, and $\mathrm{Var}[\|\epsilon_t\|^2] \leq \kappa_2 I$..*

**Theorem 1** (Validity). *Under the assumptions stated above, the proposed method satisfies the following conditional coverage guarantee:*

$$|\mathbb{P}(Y_{T+1} \in \mathcal{C}_{T+1}(\alpha)|X_{T+1} = x) - (1 - \alpha)| \leq (4L + 2L\sqrt{\omega} + 2)\sqrt{\omega} + 6\sqrt{\frac{\log(16n)}{n}} + \frac{\log(16n)}{n},$$

*where*

$$\omega = \nu_n^2 r + 2r\nu_n\sqrt{(\kappa_1 + \sqrt{\kappa_2})I} + o(g(n))(\kappa_1 + \sqrt{\kappa_2})I.$$

**Remark 3.** *The finite-sample bound on the coverage gap is directly influenced by the estimation quality and the convergence rate of $A_n$, which is given by $\max(\mathcal{O}(\frac{\log n}{n}), \mathcal{O}(\nu_n), \mathcal{O}(\sqrt{g(n)}))$. In general, reducing the coverage gap requires high-accuracy estimations (i.e., a rapidly vanishing $\nu_n$) and a well-chosen nonconformity score matrix $A_n$ that converges quickly.*

Theorem 1 highlights the importance of incorporating topology-based estimators in STACI. Relying solely on the sample covariance matrix often leads to coverage gaps in finite samples, undermining validity. In contrast, the topology-based matrix acts as a covariance estimator with topology-informed regularization, generally achieving faster convergence than the sample covariance estimator. A hybrid approach that combines both estimators provides an optimal trade-off between validity and efficiency.

## 5.2 Prediction Efficiency

We evaluate the efficiency of STACI via the volume of the prediction set in $I$-dimensional space, defined as $V(A, r) = \frac{\pi^{I/2}}{\Gamma(\frac{I}{2}+1)} \cdot r^{I/2} \cdot \det(A)^{-1/2}$. The radius of the prediction set is determined by the $(1 - \alpha)$-th quantile of the empirical CDF, computed from $n$ data points in the calibration dataset. This radius is denoted as $\hat{Q}_{1-\alpha}(\{\hat{\epsilon}_t^\top A\hat{\epsilon}_t, t \in \mathcal{D}_{\text{cal}}\})$. In the ideal case where $\hat{f}(X_t) = f(X_t)$ and $\hat{\epsilon}_t = \epsilon_t$, minimizing inefficiency reduces to: $\min_{A \succ 0} V(A, \hat{Q}_{1-\alpha}(\{\epsilon_t^\top A\epsilon_t, t \in [n]\}))$. To simplify computation, we approximate the empirical quantile with the true quantile, justified by the Glivenko-Cantelli Theorem [41], which ensures the convergence $\lim_{n\to\infty} \hat{Q}_{1-\alpha}(\{\epsilon_t^\top A\epsilon_t, t \in [n]\}) = Q_{1-\alpha}(\epsilon^\top A\epsilon)$, where $Q_{1-\alpha}(\cdot)$, the $1 - \alpha$ quantile of $\epsilon^\top A\epsilon$ is assumed to be continuous.

In the limiting case, we formulate the following minimization problem, presented in Theorem 2, and use its solution as the guiding criterion for selecting the matrix $A$:

**Theorem 2** (Efficiency). *There exists $0 < \alpha_0 < 1$, such that when $\alpha < \alpha_0$, the optimal solution to the minimization problem is given by:*

$$A_* := \arg\min_{A \succ 0} V(A, Q_{1-\alpha}(\epsilon^\top A\epsilon)), \tag{7}$$

*where $\epsilon \sim \mathcal{N}(0, A_*^{-1})$.*

**Remark 4.** *The optimization problem is invariant to scalar rescaling (i.e., $A$ and any positive scalar multiple $cA$, where $c > 0$, yield the same mathematical solution). Empirically, we find $\alpha_0 > 0.2$ for Gaussian noises when $I \leq 30$, ensuring that the result is applicable to conformal prediction settings with typical coverage levels of $90\%$ or $95\%$ in the focused setting. The analysis can go beyond the Gaussian noise assumption for $\epsilon$ and be extended to broader distributions that satisfy appropriate tail-bound conditions.*

Theorem 2 underscores the importance of selecting $A$ optimally in eq. (2) and highlights that accurately estimating the inverse of the error covariance matrix is key to minimizing inefficiency in CP. In practice, designing an optimal $A_*$ is often challenging due to empirical limitations. For example, the estimated residuals $\hat{\epsilon}_t$ may deviate significantly from the true errors $\epsilon_t$. Additionally, when the sample size $n$ is small, the empirical CDF may differ considerably from the true CDF. Despite these challenges, constructing $A$ based on an estimate of the inverse covariance matrix offers substantial improvements in high-dimensional settings compared to CP methods that ignore variable dependencies, such as those that set $A$ as the identity matrix.

## 6 Experiments

To demonstrate the suitability of STACI, we evaluate its performance on both synthetic data with a stationary covariance matrix and real-world data with time-varying covariance. By default, the first 60% of observations are used for training, the calibration set consists of the most recent $n = 500$ observations, and the test contains the sequentially revealed observations $n' = 5000$ in simulation and $n' = 5000$ in real study. The desired confidence rate $\alpha$ is fixed at 0.95. Our method is compared

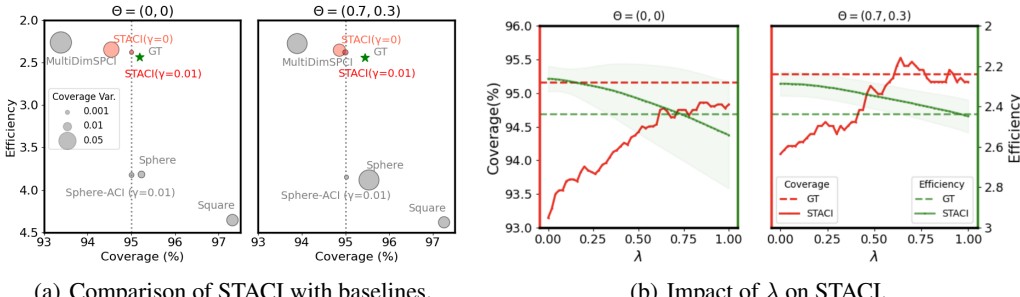

(a) Comparison of STACI with baselines.

(b) Impact of $\lambda$ on STACI.

Figure 2: Experiment results of synthetic data: (a) Comparison of methods on synthetic datasets with different tail-up parameters $\Theta$ over coverage ($x$-axis) and efficiency ($y$-axis). (b) Trade-off between coverage and efficiency on synthetic data, where the higher the better performance.

against the following conformal prediction and learning-based uncertainty quantification baselines: ($i$) **Sphere**: Spherical confidence set, where the covariance matrix is an identity matrix. In other words, the prediction error at different locations are not considered to have correlations. ($ii$) **Sphere-ACI** ($\gamma = 0.01$): Spherical confidence set with adaptive conformal inference (ACI). ($iii$) **Square**: Square confidence set. This equals to computing different nonconformity scores for each dimension, and then calibrate accordingly. ($iv$) **GT**: Ellipsoidal confidence set using the ground-truth covariance matrix. ($v$) **MultiDimSPCI**: Ellipsoidal confidence set using the sample covariance matrix [50], alongside its localized variant using the most recent observations, **MultiDimSPCI (local)**. ($vi$) **CopulaCPTS**: Prediction region based on modeling the joint distribution of forecast errors with a copula function [39]. ($vii$) **DeepSTUQ**: A Bayesian deep learning model that quantifies uncertainty in spatio-temporal graphs by using graph convolutions and Monte Carlo dropout [33].

We consider both validity and efficiency to evaluate the uncertainty quantification performance: ($i$) *Coverage* quantifies the likelihood that the prediction set includes the true target, *i.e.*, Coverage := $\sum_{t \in \mathcal{D}_{\text{test}}} \mathbb{1}\{Y_t \notin \mathcal{C}(X_t; \alpha_t)\}/n'$; ($ii$) *Efficiency* is evaluated based on the size (or volume) of the prediction set, with smaller sets indicating higher efficiency. The volume of the prediction set, $\text{Vol}(\mathcal{C}(X_t; \alpha_t))$, is measured by the size of the ellipsoid determined by $A$. Formally, Efficiency := $\sum_{t \in \mathcal{D}_{\text{test}}} \left( \text{Vol}(\mathcal{C}(X_t; \alpha_t)) \right)^{1/I}/n'$. An optimal method should achieve the predefined coverage with high efficiency/low inefficiency.

## 6.1 Simulation

In this section, we conduct simulation experiments on synthetic data generated by a tail-up model. Specifically, we follow [31] and construct the stream network as shown in Figure 1. The details of synthetic network is provided in Section D.1 of the Appendix. We generate the observation of site $u$ at time point $t$ by simulating stochastic integration from all upstream points $r \in \wedge u$ to downstream point $u$ according to eq. (4), where we set $\mu_t(u) = \sum_{i=1}^{w} \theta_i Y_{t-i}(u)$ following the AR($w$) structure and $m(\Delta) = \exp(-\Delta)$ as the exponential moving average function. The process is repeated until 5000 time steps. This experiment simulates the stream network data without any misspecification.

**Experiment Configuration** In synthetic data, the prediction model $f$ is simply a linear regression model. We first estimate parameters of in AR($w$) structure, i.e., $\Theta = (\theta_i)_{i \in [w]}$, through linear regression and then parameters in eq. (29), $\phi$ and $\sigma^2$ through $\ell_1$-loss. Parameters of $\Theta = (0, 0)$ and $\Theta = (0.7, 0.3)$ are selected for data generation. When $\Theta = (0, 0)$, the observations consist of pure noise, thus stationary; when $\Theta = (0.7, 0.3)$, the process resembles a second-order autoregressive model. The weighting factor $\lambda$ is set to 0.6.

**Results** Our numerical results demonstrate that our method enhance the predictive efficiency significantly without sacrificing the coverage guarantee, by considering both sample-based and topology-based covariance. From Figure 2(a), we observe that CP methods employing ellipsoidal uncertainty sets tend to cluster towards the upper region of the plot, indicating higher efficiency compared to CP methods based on spherical or square uncertainty sets. Although MultiDimSPCI achieves the lowest efficiency, its coverage drops significantly below the required threshold, highlighting its instability when relying solely on the sample covariance matrix. This issue persists even

Table 1: Comparison of Different CP Methods over Different GNN Models for two PeMS Datasets. Coverage below 94.5% is italicized. Inefficiency is reported as mean ± standard deviation, with the lowest value in bold.

| Dataset | Backbone Models | AGCRN | | ASTGCN | | STGODE | |
|---|---|---|---|---|---|---|---|
| | CP Methods | Coverage | Efficiency ↓ | Coverage | Efficiency ↓ | Coverage | Efficiency ↓ |
| PeMS-G1 | Sphere | 97.76 | $133.60 \pm 12.79$ | 96.64 | $128.23 \pm 14.2$ | 96.09 | $145.40 \pm 7.79$ |
| | Sphere-ACI ($\gamma = 0.01$) | 95.26 | $108.93 \pm 15.96$ | 95.24 | $114.64 \pm 19.54$ | 95.03 | $136.36 \pm 19.47$ |
| | Square | 95.98 | $155.84 \pm 23.92$ | 96.41 | $155.62 \pm 23.74$ | 96.38 | $172.30 \pm 23.51$ |
| | *MultiDimSPCI* | *92.92* | *$73.82 \pm 8.16$* | *93.19* | *$74.68 \pm 7.67$* | *92.70* | *$88.48 \pm 4.26$* |
| | *MultiDimSPCI (local)* | 93.04 | $74.23 \pm 8.27$ | 93.57 | $75.18 \pm 7.69$ | 92.98 | $88.99 \pm 4.26$ |
| | **STACI** ($\gamma = 0$) | 97.12 | $88.27 \pm 21.73$ | 96.76 | $88.1 \pm 19.07$ | 95.31 | $77.34 \pm 7.09$ |
| | **STACI** ($\gamma = 0.01$) | 95.75 | **$67.54 \pm 10.94$** | 95.54 | **$67.92 \pm 10.22$** | 95.14 | **$73.62 \pm 9.83$** |
| PeMS-G2 | Sphere | 96.26 | $144.55 \pm 14.22$ | 95.64 | $130.69 \pm 11.55$ | 95.83 | $147.13 \pm 14.68$ |
| | Sphere-ACI ($\gamma = 0.01$) | 95.03 | $133.63 \pm 14.18$ | 95.07 | $122.72 \pm 16.67$ | 95.14 | $122.34 \pm 19.72$ |
| | Square | 95.26 | $172.18 \pm 19.43$ | 95.37 | $160.60 \pm 19.95$ | 95.12 | $138.97 \pm 17.78$ |
| | *MultiDimSPCI* | *91.14* | *$101.42 \pm 7.71$* | *90.93* | *$92.12 \pm 6.67$* | *90.74* | *$107.23 \pm 7.34$* |
| | *MultiDimSPCI (local)* | 91.44 | $102.25 \pm 7.90$ | 91.12 | $92.84 \pm 6.95$ | 91.10 | $107.96 \pm 7.58$ |
| | **STACI**($\gamma = 0$) | 95.39 | $73.36 \pm 10.35$ | 95.18 | $60.45 \pm 9.22$ | 95.33 | $77.14 \pm 8.65$ |
| | **STACI**($\gamma = 0.01$) | 95.01 | **$69.62 \pm 12.85$** | 95.05 | **$58.07 \pm 9.83$** | 95.20 | **$74.86 \pm 9.63$** |

in simulated data designed to align with its error process assumptions. Its local variant, designed to capture temporal correlations, provides only a marginal improvement and similarly fails to achieve the required coverage. In contrast, with $\lambda$ fixed at $0.6$, our method STACI is positioned near the upper-right corner alongside GT, which leverages the ground-truth covariance matrix. This suggests that our method achieves performance comparable to GT, balancing low efficiency (smaller volume) while maintaining the necessary coverage guarantees. Among all methods that surpass the coverage threshold, our method, STACI($\gamma = 0.01$), demonstrates the best efficiency with the smallest variance, further reinforcing its robustness and effectiveness.

Figure 2(b) reveals a clear trade-off trend between coverage and efficiency: the higher $\lambda$, the confidence level rises, but efficiency is worse. This suggests that $\lambda$ should be carefully chosen: if too large, our method over-relies on topology and fails to adapt to covariance shift; if too small, it depends more on sample covariance matrices, which are purely data-driven and thus unstable, leading to a coverage drop. Nonetheless, no matter whether adapting confidence level, setting a larger $\lambda$ in STACI can efficiently increase coverage and maintain it near the pre-determined level, while only slightly increasing volume, which remains comparable to GT.

## 6.2 Real Data Study

We further conduct experiments on a real-world traffic dataset, Performance Measurement System (PeMS) [6], which contains the data collected from the California highway network, providing 5-minute interval traffic flow counts by multiple sensors, along with flow directions and distances between sensors. To model it into stream network, we also rely on [4] to check accurate road connection information. We select two highway forks, each equipped with 12 sensors, named PeMS-G1 and PeMS-G2, and plot their locations and corresponding road segments in Figure 5 in the Appendix.

**Experiment Configuration** We adopt Adaptive Graph Convolutional Recurrent Network (AGCRN) [2] as the default backbone model $f$. To demonstrate STACI's generality under post-hoc conformal prediction framework, we evaluate over alternative GNN backbones, including attention-based ASTGCN [16] and continuous-time STGODE [13]. We set our default $\lambda = 0.6$. For simplicity, we only use fixed weights with all equal values, without requiring any additional information. Multiple hyperparameter and ablation study are also provided over the key parameters in our framework: $(i)$ $\lambda$ from 0 to 1 with step of $0.02$; $(ii)$ $n = 100, 200, 300, 400, 500$; $(iii)$ $\gamma = 0$ or $0.01$.

**Result** Table 1 clearly demonstrates that our method consistently outperforms all baseline CP techniques across diverse backbone models and road topologies, achieving significantly higher coverage and superior efficiency. Further comparisons against CopulaTSCP and DeepSTUQ are detailed in Appendix D.5, as those methods are less fitted for our problem setting.

Our method exhibits robustness to the choice of the hyperparameter $\lambda$. In the first two subplots of Figure 3, setting $\lambda = 0$ reduces our algorithm to sole usage of sample covariance matrix, equivalent to our strongest baseline, MultiDimSPCI. Our coverage is greater than MultiDimSPCI with arbitrary hyperparameter $\lambda$. Further, across all calibration sample sizes $n$, selecting any $\lambda \in [0.3, 0.9]$ consistently yields both higher coverage and greater(lower) efficiency. This underscores the critical contribution of topological information and demonstrates STACI's insensitivity to $\lambda$.

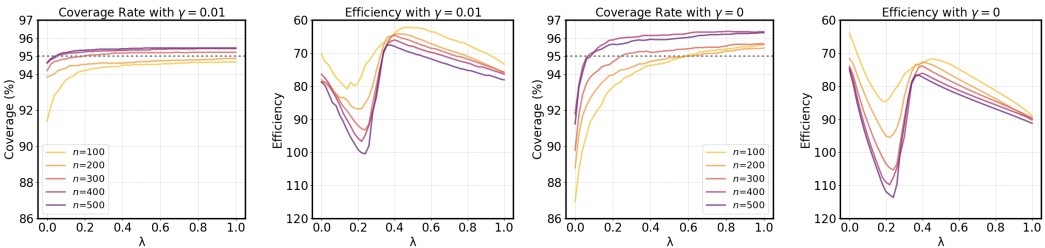

Figure 3: Comparison of Coverage and Efficiency for PeMS data with different belief weight $\lambda$ and calibration set size $n$, with adaptive step size $\gamma = 0.01$ (upper) and 0 (lower). The pre-determined coverage threshold of 95% is shown by a horizontal gray dotted line.

The incorporation of topology-induced covariance matrix is pivotal even without ACI ($\gamma = 0$), as shown in the last two subplots in Figure 3. STACI can obviously lift coverage from under 87% to surpass the desired 95% level with only a marginal efficiency cost. This indicates that under inherent temporal covariance shifts, utilizing topological information offers a robust remedy to under-coverage problem while maintaining highly informative predictions. Moreover, as demonstrated in Section D.4 in the Appendix, STACI can also outperform in an offline setting, where the covariance matrix is estimated once and then held fixed at the beginning of test time. We discuss the scalability of STACI in term of time length and graph size in Section 7 and in Appendix D.7, respectively.

In conclusion, the estimate of the covariance matrix can benefit from both topology and samples, compared with relying on any single resource. On one hand, with limited finite calibration sample $n$, the topology-based estimator offers a more stable structure as it possess fewer parameters. It can alleviate the temporal distribution shift and the resulting under-cover problem, consistent with our theoretical analysis. The sample covariance, on the other hand, captures the actual spatial patterns from samples in the calibration data and gives higher efficiency, but it can lose coverage guarantee if the distribution changes. By blending these two estimates and adaptively adjusting the significance level, STACI can effectively maintain desired coverage and smaller volumes.

## 7 Limitation

STACI targets joint uncertainty quantification on *moderate*-size subgraphs ($2 \sim 30D$). This choice is deliberate: performing UQ in very high-dimensional output spaces (hundreds of coordinates or more) is statistically ill-posed. Directly constructing geometric prediction sets in the full output space is vulnerable to the curse of dimensionality [44]. In particular, hyper-rectangles formed by marginal intervals can inflate to near-vacuous volumes even at low dimensions as in Table 1.

In high-dimensional regimes, common practice is to avoid explicit geometric sets and instead use *generative* or *sampling*-based UQ [26, 3]. For instance, image UQ, a prototypical high-dimensional setting, typically samples plausible outcomes from a learned posterior [12] rather than delineating a set in pixel space. In the same spirit, STACI focuses on subgraphs of manageable joint dimensionality so that (i) the resulting joint sets remain informative, (ii) the coverage–efficiency trade-off is non-degenerate. This design also matches how practitioners assess risk: traffic engineers often analyze corridors with 10–30 sensors rather than entire city-wide networks, and hydrologists frequently study clusters of ~20 gauges when evaluating localized flood risk.

## 8 Conclusion

We proposed STACI, an adaptive conformal prediction framework for stream networks. Theoretically, we established coverage guarantees and demonstrated the model's ability to minimize inefficiency under mild conditions. Empirically, STACI produced smaller prediction sets while maintaining valid coverage across both (stationary) simulated data and (non-stationary) real-world traffic data.

Future work includes three potential directions. (i) STACI can be extended to general spatio-temporal graphs by replacing $\Sigma_{\mathcal{G}}$ with alternative network parameterizations, enabling the development of novel methods that effectively exploit topological structures in broader spatio-temporal settings; (ii) stronger theoretical guarantees such as finite-sample coverage bounds when adaptively calibrating the significance level, could be developed by imposing assumptions on error distribution shifts (e.g., first-order differencing stationarity); and (iii) the present formulation does not aim to provide full-graph joint sets over hundreds of nodes. One might build corridor-level joint sets and composing them with rigorous controls on cross-correlation to scale coverage beyond the subgraph level.

## Acknowledgements

We thank Aravindan Vijayaraghavan, Shuwen Chai, Yifan Wu for helpful discussions. We also thank anonymous reviewers for constructive feedback.

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

# Contents

# A  Theoretical proofs

## A.1  Proof of Theorem 1

**Theorem 1**(validity). Under the assumptions stated above, the proposed method satisfies the following conditional coverage guarantee:

$$|\mathbb{P}(Y_{T+1} \in \mathcal{C}_{T+1}(\alpha)|X_{T+1} = x) - (1 - \alpha)| \leq (4L + 2L\sqrt{\omega} + 2)\sqrt{\omega} + 6\sqrt{\frac{\log(16n)}{n}} + \frac{\log(16n)}{n},$$

where

$$\omega = \nu_n^2 r + 2r\nu_n\sqrt{(\kappa_1 + \sqrt{\kappa_2})I} + o(g(n))(\kappa_1 + \sqrt{\kappa_2})I.$$

For easy notation, denote $\hat{A} = A_n, \Delta_t = \hat{\epsilon}_t - \epsilon_t$ and sometimes we drop subscript $t$.

**Lemma 1.** *For any test conformity score $\hat{s}_t = \hat{\epsilon}_t^T \hat{A}\hat{\epsilon}_t$ and the true conformity score $s_t = \epsilon_t^T A\epsilon_t$, with probability at least $1 - \delta$,*

$$\sum_{t=T-n+1}^{T} |\hat{s}_t - s_t| \leq \omega n, \tag{8}$$

*where*

$$\omega = \nu_n^2 r + 2r\nu_n\sqrt{(\kappa_1 + \sqrt{\kappa_2})I} + o(g(n))(\kappa_1 + \sqrt{\kappa_2})I.$$

*Proof.* We have:

$$|\hat{s}_t - s_t| = |\epsilon_t^\top A\epsilon_t - \hat{\epsilon}_t^\top \hat{A}\hat{\epsilon}_t| \leq |\epsilon_t^\top A\epsilon_t - \epsilon_t^\top \hat{A}\hat{\epsilon}_t| + |\epsilon_t^\top \hat{A}\epsilon_t - \hat{\epsilon}_t^\top \hat{A}\hat{\epsilon}_t|$$

$$\leq |\Delta^\top \hat{A}\Delta| + 2|\Delta^\top \hat{A}\epsilon_t| + |\epsilon_t^\top (A - \hat{A})\epsilon_t|$$

$$\leq \|\hat{A}\|\|\Delta\|^2 + 2\|\hat{A}\|\|\Delta\|\|\epsilon\| + \|\epsilon\|^2\|A - \hat{A}\| \tag{i}$$

$$\leq r\|\Delta\|^2 + 2r\|\Delta\|\|\epsilon\| + o(g(n))\|\epsilon\|^2. \tag{9}$$

The inequality (i) exists because of the Cauchy-schwartz inequality and Assumption 2. Hence, by Assumption 1, we have

$$\sum_{t=T-n+1}^{T} |\hat{s}_t - s_t| \leq rn\nu_n^2 + 2r\sum_{t=T-n+1}^{T} \|\Delta_t\|\|\epsilon_t\| + o(g(n))\sum_{t=T-n+1}^{T} \|\epsilon_t\|^2$$

$$\leq rn\nu_n^2 + 2r\sqrt{(\sum_{t=T-n+1}^{T} \|\Delta_t\|^2)(\sum_{t=T-n+1}^{T} \|\epsilon_t\|^2)} + o(g(n))\sum_{t=T-n+1}^{T} \|\epsilon_t\|^2$$

$$\leq rn\nu_n^2 + 2r\sqrt{n\nu_n^2(\sum_{t=T-n+1}^{T} \|\epsilon_t\|^2)} + o(g(n))\sum_{t=T-n+1}^{T} \|\epsilon_t\|^2. \tag{10}$$

From Assumption 3, we have

$$\mathbb{E}\left[\frac{1}{n}\sum_{t=T-n+1}^{T} \|\epsilon_t\|^2\right] = \frac{1}{n}\sum_{t=T-n+1}^{T} \mathbb{E}[\|\epsilon_t\|^2] \leq \kappa_1 I. \tag{11}$$

Using Chebyshev's inequality, we have

$$\mathbb{P}\left(\frac{1}{n}\sum_{t=T-n+1}^{T} \|\epsilon_t\|^2 - \mathbb{E}[\|\epsilon_t\|^2] \geq \sqrt{\frac{\text{Var}[\|\epsilon_t\|^2]}{n\delta}}\right) \leq \frac{\text{Var}[\|\epsilon_t\|^2]}{n \cdot \frac{\text{Var}[\|\epsilon_t\|^2]}{n\delta}} = \delta, \tag{12}$$

which means that with probability higher than $1 - \delta$,

$$\frac{1}{n} \sum_{t=T-n+1}^{T} \|\epsilon_t\|^2 \leq \mathbb{E}[\|\epsilon_t\|^2] + \sqrt{\frac{\mathrm{Var}[\|\epsilon_t\|^2]}{n\delta}}$$

$$\leq \kappa_1 I + \sqrt{\frac{\kappa_2 I}{n\delta}}$$

$$\leq (\kappa_1 + \sqrt{\frac{\kappa_2}{n\delta I}})I \leq (\kappa_1 + \sqrt{\kappa_2})I. \tag{13}$$

The last inequality is because we can set $\delta$ such that $\delta n I < 1$. Plug into eq. (10), we have with probability higher than $1 - \delta$, we obtain eq. (8) and the lemma follows.

$\square$

Denote the empirical CDF: $\widehat{F}_{n+1}(x) = \frac{1}{n} \sum_{i=T-n+1}^{T} 1_{\hat{s}_i \leq x}$, $\widetilde{F}_{n+1}(x) = \frac{1}{n} \sum_{i=T-n+1}^{T} 1_{s_i \leq x}$ and true CDF of score function $F_s(x) = P(s \leq x)$.

**Lemma 2.** *Under Assumption 3, for any $n$, there exists an event $A_n$ which occurs with probability at least $1 - \sqrt{\frac{\log(16n)}{n}}$, such that, conditioning on $A_n$,*

$$\sup_x \left| \widetilde{F}_{n+1}(x) - F(x) \right| \leq \sqrt{\frac{\log(16n)}{n}}.$$

*Proof.* The proof follows Lemma 1 in [51] that utilizes Dvoretzky-Kiefer-Wolfowitz inequality in [24]. $\square$

**Lemma 3.** *Under Assumption 1, Assumption 3,with high probability,*

$$\sup_x \left| \widehat{F}_{n+1}(x) - \widetilde{F}_{n+1}(x) \right| \leq (2L+1)\sqrt{\omega} + 2 \sup_x \left| \widetilde{F}_{n+1}(x) - F_e(x) \right|.$$

*Proof.* The proof is similar to Lemma $B.6$ in [50], and is written here for completeness.

Using Lemma 1 we have that with probability $1 - \delta$,

$$\sum_{t=T-n+1}^{T} |s_t - \hat{s}_t| \leq n\omega. \tag{14}$$

Let $S = \{t : |s_t - \hat{s}_t| \geq \sqrt{\omega}\}$. Then,

$$|S|\sqrt{\omega} \leq \sum_{t=T-n+1}^{T} |s_t - \hat{s}_t| \leq n\omega. \tag{15}$$

So $|S| \le n\sqrt{\omega}$. Then,

$$
\begin{aligned}
|\widehat{F}_{n+1}(x) - \widetilde{F}_{n+1}(x)| &\le \frac{1}{n} \sum_{t=T-n+1}^{T} |1\{\hat{s}_t \le x\} - 1\{s_t \le x\}| \\
&\le \frac{1}{n}|S| + \sum_{t \notin S} |1\{\hat{s}_t \le x\} - 1\{s_t \le x\}| \\
&\le \frac{1}{n}|S| + \frac{1}{n} \sum_{t=T-n+1}^{T} 1\{|s_t - x| \le \sqrt{\omega}\} \quad\quad\quad\quad\quad\quad (i) \\
&\le \sqrt{\omega} + P(|s_{T+1} - x| \le \sqrt{\omega}) \\
&\quad + \sup_x \left| \frac{1}{n} \sum_{t=T-n+1}^{T} 1\{|s_t - x| \le \sqrt{\omega}\} - P(|s_{T+1} - x| \le \sqrt{\omega}) \right| \\
&= \sqrt{\omega} + [F_s(x + \sqrt{\omega}) - F_s(x - \sqrt{\omega})] \\
&\quad + \sup_x \left[ \widetilde{F}_{n+1}(x + \sqrt{\omega}) - \widetilde{F}_{n+1}(x - \sqrt{\omega}) - \left( F_s(x + \sqrt{\omega}) - F_s(x - \sqrt{\omega}) \right) \right]
\end{aligned}
$$
$$(ii)$$
$$
\le (2L + 1)\sqrt{\omega} + 2 \sup_x \left| \widetilde{F}_{n+1}(x) - F_s(x) \right|, \tag{16}
$$

where $(i)$ is because $|1\{a \le x\} - 1\{b \le x\}| \le 1\{|b - x| \le |a - b|\}$ for $a, b \in \mathbb{R}$, and $(ii)$ is due to the Lipschitz continuity of $F_s(x)$. $\qquad\square$

**Proof of Theorem 1**

*Proof.* Look at the conditional coverage of $Y_{T+1}$ given $X_{T+1}$:

$$
\begin{aligned}
&\left| \mathbb{P}\left( Y_{T+1} \in \mathcal{C}_{T+1}^{\alpha} \mid X_{T+1} = x_{T+1} \right) - (1 - \alpha) \right| \tag{17} \\
&= \left| \mathbb{P}\left( \hat{s}_{T+1} \le 1 - \alpha \text{ quantile of } \widehat{F}_{n+1} \mid X_{T+1} = x \right) - (1 - \alpha) \right| \\
&= \left| \mathbb{P}\left( \widehat{F}_{n+1}(\hat{s}_{T+1}) \le 1 - \alpha \right) - \mathbb{P}\left( F_s(s_{T+1}) \le 1 - \alpha \right) \right| \\
&= \left| \mathbb{E}[1\{\widehat{F}_{n+1}(\hat{s}_{T+1}) \le 1 - \alpha\} - 1\{F_s(s_{T+1}) \le 1 - \alpha\}] \right| \\
&\le \mathbb{P}\left( |F_s(s_{T+1}) - (1 - \alpha)| \le |\widehat{F}_{n+1}(\hat{s}_{T+1}) - F_s(s_{T+1})| \right). \tag{18}
\end{aligned}
$$

Based on Lemma 2, we can define the event $A_n$, $\mathbb{P}(A_n) \ge 1 - \frac{\log(16n)}{n}$, conditional on $A_n$, we have:

$$
\sup_x \left| \widetilde{F}_{n+1}(x) - F_s(x) \right| \le \sqrt{\frac{\log(16n)}{n}}, \tag{19}
$$

Hence, we can write eq. (18) as

$$
\begin{aligned}
&\mathbb{P}(|F_s(s_{T+1}) - (1 - \alpha)| \le |\widehat{F}_{n+1}(\hat{s}_{T+1}) - F_s(s_{T+1})|) \\
\le& \mathbb{P}(|F_s(s_{T+1}) - (1 - \alpha)| \le |\widehat{F}_{n+1}(\hat{s}_{T+1}) - F_s(s_{T+1})||A_n) + \mathbb{P}(A_n^c) \\
\le& \mathbb{P}(|F_s(s_{T+1}) - (1 - \alpha)| \le |\widehat{F}_{n+1}(\hat{s}_{T+1}) - F_s(\hat{s}_{T+1})| + |F_s(\hat{s}_{T+1}) - F_s(s_{T+1})||A_n) + \frac{\log(16n)}{n}.
\end{aligned}
$$
$$(20)$$

Conditional on $A_n$:

$$
\begin{aligned}
&|\widehat{F}_{n+1}(\hat{s}_{T+1}) - F_s(\hat{s}_{T+1})| + |F_s(\hat{s}_{T+1}) - F_s(s_{T+1})| \\
\le& \sup_x |\widehat{F}_{n+1}(x) - F_s(x)| + L|\hat{s}_{T+1} - s_{T+1}| \\
\le& (2L + 1)\sqrt{\omega} + 3\sqrt{\frac{\log(16n)}{n}} + L\omega. \tag{21}
\end{aligned}
$$

The last equation exists because of Lemma 3 1 and eq. (19).

Note that $F_s(s_{T+1}) \sim Unif(0, 1)$, we have

$$\mathbb{P}(|F_s(s_{T+1}) - (1 - \alpha)| \leq |\widehat{F}_{n+1}(\widehat{s}_{T+1}) - F_s(\widehat{s}_{T+1})| + |F_s(\widehat{s}_{T+1}) - F_s(s_{T+1})||A_n)$$

$$\leq (4L + 2)\sqrt{\omega} + 6\sqrt{\frac{\log(16n)}{n}} + 2L\omega. \tag{22}$$

Plug into eq. (18), we have

$$\left| \mathbb{P}\left( Y_{T+1} \in \widehat{C}_{T+1}^\alpha \mid X_{T+1} = x_{T+1} \right) - (1 - \alpha) \right|$$

$$\leq (4L + 2)\sqrt{\omega} + 6\sqrt{\frac{\log(16n)}{n}} + 2L\omega + \frac{\log(16n)}{n}. \tag{23}$$

$\square$

## A.2 Proof of Theorem 2

**Theorem 2** (Efficiency). There exists $0 < \alpha_0 < 1$, such that when $\alpha < \alpha_0$, the optimal solution to the minimization problem is given by:

$$A_* := \arg\min_{A \succ 0} V(A, Q_{1-\alpha}(\epsilon^\top A\epsilon)), \tag{24}$$

where $\epsilon \sim \mathcal{N}(0, A_*^{-1})$.

**Lemma 4** (Uniform tail threshold via Dirichlet–Jensen). *Let $S_\lambda = \sum_{i=1}^I \lambda_i Z_i^2$ with $Z_i \overset{i.i.d}{\sim} \mathcal{N}(0, 1)$ and $\lambda \in \Delta^{I-1}$. Then $S_\lambda \overset{d}{=} TU_\lambda$ with $T \sim \chi_I^2$ independent of $U_\lambda = \sum_i \lambda_i V_i$, where $V \sim$ Dirichlet$(\frac{1}{2}, \ldots, \frac{1}{2})$. For all $\lambda$ we have $0 < U_\lambda \leq 1$ and $\mathbb{E}[U_\lambda] = 1/I$. Moreover, for $s \geq I + 2$, the map $u \mapsto \bar{F}_I(s/u)$ is convex on $(0, 1]$ (where $\bar{F}_I$ is the $\chi_I^2$ survival function). Consequently,*

$$\mathbb{P}(S_\lambda \geq s) = \mathbb{E}[\bar{F}_I(s/U_\lambda)] \geq \bar{F}_I(s/\mathbb{E}[U_\lambda]) = \bar{F}_I(Is) = \mathbb{P}(S_{\lambda^\star} \geq s).$$

*Equivalently, with $p^*(I) := F_{\chi_I^2}(I(I + 2)) \in (0, 1)$, we have*

$$Q_p(S_{\lambda^\star}) \leq Q_p(S_\lambda), \qquad \forall \lambda \in \Delta^{I-1}, \forall p > p^*(I).$$

*Proof.* The Dirichlet–$\chi^2$ factorization and $\mathbb{E}[U_\lambda] = 1/I$ are standard. Write $g_s(u) := \bar{F}_I(s/u)$. Using $\chi_I^2$ density $f_I(x) \propto x^{I/2-1}e^{-x/2}$, one computes $g_s''(u) \geq 0$ iff $f_I'(x) \leq -\frac{2}{x}f_I(x)$ at $x = s/u$. Since $f_I'/f_I = (\frac{I}{2} - 1)\frac{1}{x} - \frac{1}{2}$, the inequality holds whenever $x \geq I + 2$. If $s \geq I + 2$ then $x = s/u \geq s \geq I + 2$ for all $u \in (0, 1]$, hence $g_s$ is convex on $(0, 1]$. Apply Jensen and note $S_{\lambda^\star} \overset{d}{=} \frac{1}{I}\chi_I^2$ to conclude. $\square$

## Proof of Theorem 2

*Proof.* By the definition of ellipsoid volume, we have

$$V(A, Q_{1-\alpha}(\epsilon^\top A\epsilon)) = Constant \times Q_{1-\alpha}(\epsilon^\top A\epsilon)^{I/2}[\det(A)]^{-1/2} \tag{25}$$

Since the optimization problem is invariant to the rescaling of the scalar (that is, $A$ and any positive scalar multiple $cA$, where $c > 0$, produce the same mathematical solution), additional constraints, such as the bounding of the matrix norm $A \leq 1$, can be imposed without loss of generality. Note that $\epsilon \sim N(0, A_*^{-1})$ and $A \succ 0$ by Cholesky decomposition, we can write $A_*^{-1} = LL^\top$ where $L$ is a lower triangular matrix. Define the matrix $B = L^\top AL$, we can rewrite the eq. (25) as

$$\min_{B \succ 0} Q_{1-\alpha}^{I/2}(x^\top Bx) \det(L)[\det(B)]^{-1/2}, \tag{26}$$

where $x \sim N(0, \text{Id})$ and $\det(L)$ is a constant independent of $B$.

To further solve the optimization problem, we look at the eigenvalue of $B$, suppose $B = O\text{diag}(\lambda_1, \lambda_2, \cdots, \lambda_I)O^\top$ and $O$ is an orthogonal matrix. Since the optimization value is invariant to different scaling of $B$, we are imposing additional constrains on $\lambda_i$.

$$\min_{\lambda_i \geq 0, \sum_{i=1}^I \lambda_i = 1} Q_{1-\alpha}^{I/2} \left( \sum_{i=1}^I \lambda_i x_i^2 \right) \prod_{i=1}^I \lambda_i^{-\frac{1}{2}}, \tag{27}$$

and $\{x_i\}_{1 \leq i \leq I}$ are i.i.d. random variables $x_i \sim N(0, 1)$.

Now we would like to prove that the above optimization problem is solved when $\lambda^* = (\frac{1}{I}, \cdots, \frac{1}{I})$. Note that by Cauchy-Schwartz Inequality $\prod_{i=1}^I \lambda_i^{-\frac{1}{2}}$ is minimized when $\lambda_1 = \lambda_2 \cdots = \lambda_d = \frac{1}{I}$, by Lemma 4, $Q_{1-\alpha}(\sum \lambda_i x_i^2)$ is minimized in $\lambda^*$, for $\alpha < \alpha_0$ and $\alpha_0$ is decided according to $I$. Theorem 2 follows.

$\square$

## A.3 Tail-up model

**Lemma 5.** *The spatial covariance $\Sigma$ of any node pair $(u, v)$ is:*

$$\Sigma(u, v) = \int_{\wedge u \cap \wedge v} m(r - u)m(r - v) \frac{w(r)}{\sqrt{w(u)w(v)}} dr. \tag{28}$$

*Proof.* Note that

$$\Sigma(u, v) = \text{Cov}(\int_{\wedge u} m(s - u)\sqrt{\frac{w(s)}{w(u)}} dB(s), \int_{\wedge v} m(r - v)\sqrt{\frac{w(r)}{w(v)}} dB(r))$$

Due to the independence of increments for Brownian motion, only when $r = s \in \wedge u \cap \wedge v$, the covariance is non-zero. Note that for Brownian motion, we have $\text{Cov}(dB(r), dB(s)) = 1_{r=s} dr ds$. Hence Lemma 5 follows. $\square$

**Lemma 6.** *If we set the moving function $m(r - u) = \beta \exp\left(-\frac{d(r,u)}{\phi}\right)$, with parameters $\beta > 0$ (a scale factor) and $\phi > 0$ (a range or decay parameter), then the covariance matrix between two locations $u, v$ can be expressed as:*

$$\Sigma(u, v) = \sigma^2 \sqrt{\frac{w(u)}{w(v)}} \exp\left(-d(u, v)/\phi\right) \mathbf{1}_{\{u \to v\}}. \tag{29}$$

*Proof.* By Lemma 5 and substitute $m(r - u) = \beta\, e^{-d(r,u)/\phi}$ and $m(r - v) = \beta\, e^{-d(r,v)/\phi}$, we have

$$\Sigma(u, v) = \int_{\wedge u \cap \wedge v} \beta^2 \exp\left(-\frac{d(r,u)}{\phi}\right) \exp\left(-\frac{d(r,v)}{\phi}\right) \frac{w(r)}{\sqrt{w(u)\, w(v)}}\, dr.$$

The set $\wedge u \cap \wedge v$ are the segments of networks that flow into *both* $u$ and $v$. Consider the following cases:

- *If $u$ and $v$ are not flow-connected*, then $\wedge u \cap \wedge v = \emptyset$ and hence $\Sigma(u, v) = 0$.

- *If $u$ and $v$ are flow-connected*, without loss of generality assume $v$ is downstream of $u$. Then $\wedge u \cap \wedge v = \wedge u$, and for each $r$ in $\wedge u$, $d(r, v) = d(r, u) + d(u, v)$. Hence we have

$$\exp\left(-\frac{d(r, u) + d(r, v)}{\phi}\right) = \exp\left(-\frac{d(u, v)}{\phi}\right) \exp\left(-\frac{2\, d(r, u)}{\phi}\right).$$

  leads to a remaining integral over $r \in \wedge u$. We can write

$$\Sigma(u, v) = \beta^2 \sqrt{\frac{w(u)}{w(v)}} \exp\left(-\frac{d(u, v)}{\phi}\right) \int_{\wedge u} \exp\left(-\frac{2d(r,u)}{\phi}\right) \frac{w(r)}{w(u)}\, dr,$$

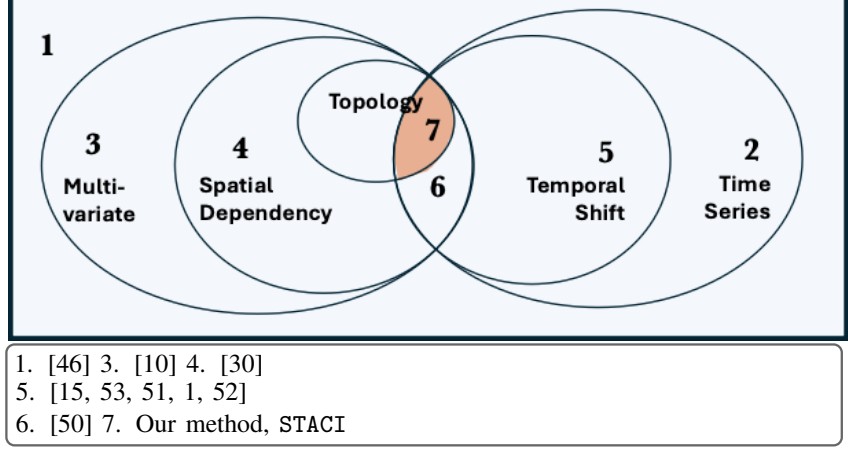

1. [46] 3. [10] 4. [30]
5. [15, 53, 51, 1, 52]
6. [50] 7. Our method, `STACI`

Figure 4: Taxonomy of works in conformal prediction. Among studies that account for both spatial dependency and temporal shift—without assuming spatial and temporal exchangeability—our work is the first to incorporate topology information.

Note that $\int_{\wedge u} \exp\left(-\frac{2d(r,u)}{\phi}\right) \frac{w(r)}{w(u)}\, dr = \Sigma(u,u)$ is a constant, since the additivity constraint on $w(u)$ assures the constant variance of site $u$. Thus, the tail-up exponential model yields a covariance of the form

$$\Sigma(u,v) = \begin{cases} \left(\text{constant factors}\right) \exp\left(-\frac{d(u,v)}{\phi}\right), & \text{if } u \text{ and } v \text{ are flow-connected,} \\ 0, & \text{otherwise.} \end{cases}$$

$\square$

## B  Taxonomy for Related Works

Figure 4 provides an overview of the conformal prediction (CP) literature, complementing the discussion in the related work section. The Venn diagram categorizes existing CP methods based on the type of data they are designed for—time series, multivariate data, or both—and the nature of the assumptions they make, particularly regarding exchangeability, temporal stationarity, and spatial independence.

Traditional CP methods, such as split conformal prediction for i.i.d. regression[46], assume full exchangeability, and thus cannot be directly applied to time series or spatially structured data without modification. In the time series domain, recent works such as [1, 53, 15, 52] relax the exchangeability assumption via online calibration, sliding window methods, scorecasters or dynamic programming. These methods handle distribution shift over time but typically operate in univariate or low-dimensional settings. In contrast, CP methods for multivariate or high-dimensional data [30, 10] often focus on constructing prediction sets that exploit geometry or sparsity, but generally assume no temporal structure.

Our proposed method, *STACI*, is positioned at the intersection of these axes in the Venn diagram. It is designed for high-dimensional time-indexed multivariate data arising from spatio-temporal stream networks. Our method explicitly accounts for both spatial dependencies and temporal shifts, leveraging the underlying topological structure of the network to enhance predictive performance.

## C  Adaptive Uncertainty Set Construction

We present the analysis of the average coverage guarantee of `STACI` without any assumption about $\epsilon_t$. The proof follows from Proposition 4.1 in [15].

**Algorithm 1** `STACI`

---

**Input:** Data $\mathcal{D}$; Network topology $\mathcal{G}$; Model $f(\cdot)$; Hyper-parameters $\lambda$; Confidence level $\alpha$.
**Output:** Prediction set $\mathcal{C}(X_{T+1}; \alpha)$.

1: // `Training`
2: $\hat{f} \leftarrow$ Fit $f$ using $\mathcal{D} \setminus \mathcal{D}_{\text{cal}}$;
3: // `Calibration`
4: $\mathcal{E} \leftarrow \{\hat{\epsilon}_t = Y_t - \hat{f}(X_t) - \frac{\sum_{t \in \mathcal{D}_{\text{cal}}}(Y_t - \hat{f}(X_t))}{|\mathcal{D}_{\text{cal}}|}\}_{t \in \mathcal{D}_{\text{cal}}}$;
5: $\hat{\Sigma}_n \leftarrow \sum_{t \in \mathcal{D}_{\text{cal}}} \hat{\epsilon}_t \hat{\epsilon}_t^\top / (n-1)$ given $\mathcal{E}$;
6: $\hat{\Sigma}_{\mathcal{G}} \leftarrow$ Compute (29) for $(\ell_i, \ell_{i'}), \forall i, i' \in \mathcal{I}$ given $\mathcal{G}$;
7: $A \leftarrow \lambda \hat{\Sigma}_{\mathcal{G}}^{-1} + (1 - \lambda) \hat{\Sigma}_n^{-1}$;
8: $\mathcal{S} \leftarrow \{\hat{\epsilon}_t^\top A \hat{\epsilon}_t\}_{t \in \mathcal{D}_{\text{cal}}}$ given $\mathcal{E}$;
9: $\hat{Q}_{1-\alpha} \leftarrow$ Compute $\frac{\lceil (1-\alpha)(n+1) \rceil}{n}$-th largest element of $\mathcal{S}$;
10: // `Testing`
11: $\mathcal{C}(X_{T+1}; \alpha) \leftarrow \{y : s(X_{T+1}, y) \leq \hat{Q}_{1-\alpha}\}$;

---

**Proposition 1.** *Consider $n'$ test data points as the $n'$ realizations of $(X_{T+1}, Y_{T+1})$, denoted by $\mathcal{D}_{test}$. We have the asymptotic coverage guarantee:*

$$\lim_{n' \to \infty} \sum_{t \in \mathcal{D}_{test}} \mathbb{1}\{Y_t \notin \mathcal{C}(X_t; \alpha_t)\}/n' = 1 - \alpha.$$

While Proposition 1 provides a weaker coverage guarantee compared to Theorem 1, it offers broader applicability, remaining valid even in adversarial online settings. Empirical results suggest that when the error process exhibits minimal distribution shift and the assumptions of Theorems 2 and 1 are only slightly violated, `STACI` maintains the predefined coverage level ($\gamma = 0.01$) while achieving efficient prediction sets. However, when $\gamma > 0$, Proposition 1 does not ensure a finite-sample coverage gap. Understanding this limitation and developing methods to control the finite-sample coverage gap presents an interesting direction for future research.

## D  Additional Experiment Details and Results

### D.1  Computational Resource

Experiments were conducted on a single NVIDIA GeForce RTX 4080 Super GPU, an AMD Ryzen 9 7950X 16-Core Processor CPU, 64GB Memory and 2TB SSD.

### D.2  Datasets and Codes

The PeMS03 dataset used in this paper is collectd by California Transportation Agencies (CalTrans) Performance Measurement System (PeMS). It contains three months of statistics on traffic flow every 5 minutes ranging from Sept. 1st 2018 to Nov. 30th 2018, including 358 sensors. The datasets are from `https://github.com/guoshnBJTU/ASTGNN/tree/main` without available license.

The first backbone ST-Graph model is AGCRN [2], which we adapted the official implementation from `https://github.com/LeiBAI/AGCRN` under MIT license. ASTGCN [16] was implemented with PyTorch Geometric Temporal package [34] from `https://github.com/benedekrozemberczki/pytorch_geometric_temporal` under MIT license. STGODE [13] was adapted from the official implementation `https://github.com/square-coder/STGODE` under Apache-2.0 license.

### D.3  Simulation Data Generation Details

Segment $r_1$ and $r_2$ starts with (0, 1) and (0.5, 0.8), respectively, and both end with (0.3, 0.5). The next segment $r_3$ also starts with (0.3, 0.5), and end with (0.2, 0.1). Segment $r_4$ start from (0.6, 0.6), and ends at the same location as $r_3$. Starting from this location, $r_5$ ends at (0.4, 0). The weights for

segment $1 - 5$ are set as $0.35, 0.5, 0.85, 0.15$ and $1$, respectively. Each segment has two observation locations – one at the start point, another at the middle point.

To approximate the integral, each segment is uniformly divided into 300 smaller sub-intervals. For segments without parent nodes ($r_1$, $r_2$ and $r_4$ in our example), the source nodes are treated as infinitely distant. In implementation, the source node of each segment is extended 10 times in the same direction to simulate infinity.

## D.4  Real-world Data Details

As shown in Figure 5, we construct two subgraphs from PeMS03 dataset. We construct PEMS03-G1 and PEMS03-G2 with temporal distribution shift shown in Table 2. Note that road network structure are required in stream networks. Among all PeMS datasets, PeMS03 is the only one that contains a mapping to real sensor identification number in PeMS. Therefore, other datasets are not available for our setting.

Table 2: Temporal distribution shift of constructed datasets

| Dataset | Intra-Period Variance | Inter-Period Flow Shift |
|---|---|---|
| PEMS03-G1 | 117.4614 | 21.5453 |
| PEMS03-G2 | 106.2113 | 17.4750 |

## D.5  Main Experiment Continued

We evaluated two additional baselines, CopulaTSCP and DeepSTUQ, on the PeMS-G1 dataset, with the results summarized in the Table 3. Using the same backbone model AGCRN, CopulaTSCP significantly underperforms our method and even general CP methods. This is because that learning the joint cumulative distribution function, a requirement for CopulaTSCP, is highly unstable and unreliable in high-dimensional settings like our 12-dimensional joint prediction task. DeepSTUQ, a Bayesian GNN method, provided better efficiency but still failed to outperform STACI. Since DeepSTUQ produces marginal intervals, we aggregated them into a hyper-rectangle for a joint comparison. However, this aggregation lacks a formal guarantee of joint coverage, and applying a formal correction like Bonferroni would result in overly conservative and inefficient prediction sets. These results underscore the limitations of existing methods in high-dimensional joint prediction and highlight the need for specialized approaches.

Table 3: Comparison over additional baselines on the PeMS-G1 dataset.

| Method | Coverage | Efficiency |
|---|---|---|
| CopulaCPTS(AGCRN) | 70.60 | 668.00 |
| DeepSTUQ | 93.82 | 66.51 |
| STACI(AGCRN) | 95.75 | 67.54 |

## D.6  Robustness under imperfect topological information

To evaluate our model's robustness against graph structure noise, we conducted an experiment by perturbing the graphs of the PeMS dataset. A "noisy" graph was constructed by adding 12 spurious edges: for each of the 12 nodes in the original graph, we randomly selected another node and added an edge between them. As shown in the Table 4, the performance of all GNN-based methods degraded when using the noisy graph. Notably, our method, STACI, demonstrated the strongest robustness to these perturbations. It experienced the least performance degradation among all models, showcasing its resilience to imperfect graph structures.

## D.7  Running Time

On a single Nvidia Geforce 4080S, the training time of AGCRN with PEMS-G1 is $1419s$. In table 5, we show the computation time of our and baseline CP methods on AGCRN model and PEMS-G1 dataset. Our method `STACI` has comparable running time with other methods. Here we provide a detailed complexity analysis. Compare to MultiDimSPCI, for each test time point we

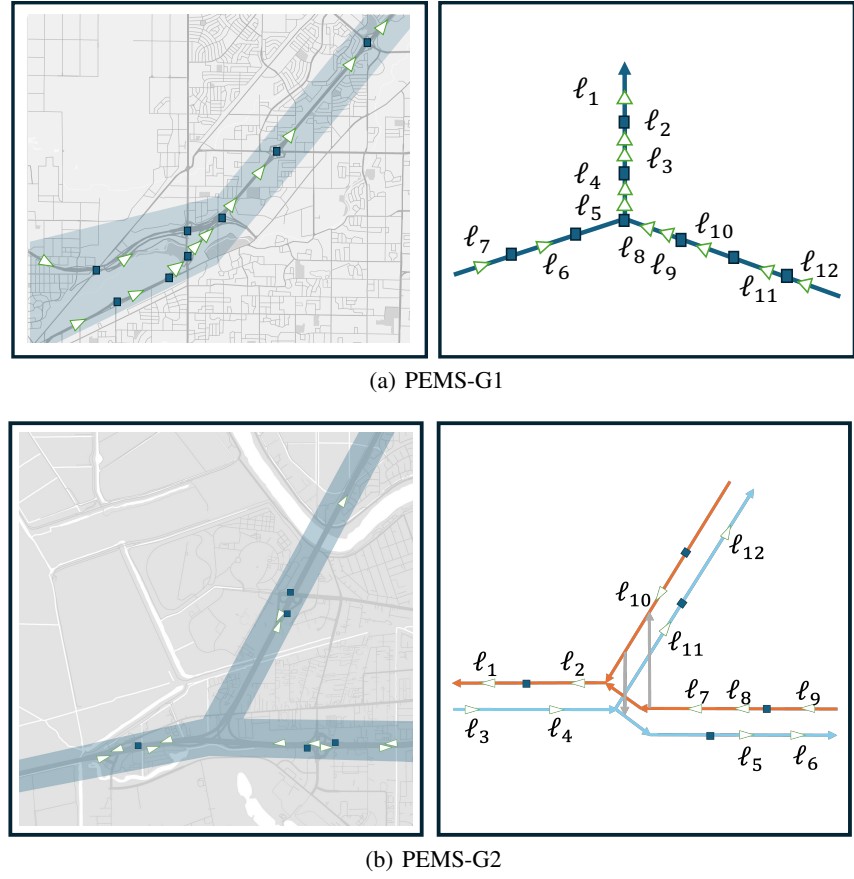

(a) PEMS-G1

(b) PEMS-G2

Figure 5: Real-world road network structures and their abstraction for PEMS-G1 and PEMS-G2. In each sub-figure, the left map displays the road network, where freeways are bold gray lines in blue shade, and ramps off the freeway are represented by blue squares. Based on these ramps and road junctions, the network is divided into different segments. Traffic flow monitoring sensors from $\ell_1$ to $\ell_{12}$ are placed exclusively on those northbound freeways, marked with green transparent triangles. The right map provides an abstract representation of the road network and sensor locations, using the same symbols for consistency.

Table 4: Robustness of STACI to noisy sensor locations on the PeMS-G1 and PeMS-G2 datasets across various GNN backbones. Noise Scale represents the standard deviation of the Gaussian noise added to location coordinates. The bolded rows indicate performance on the original, clean data.

| Dataset | Noise Scale | AGCRN | | ASTGCN | | STGODE | |
|---|---|---|---|---|---|---|---|
| | | Coverage | Efficiency | Coverage | Efficiency | Coverage | Efficiency |
| G1 | **0.0** | **95.75** | **67.54 ± 10.94** | **95.54** | **67.92 ± 10.22** | **95.14** | **73.62 ± 9.83** |
| | 0.5 | 95.83 | 71.94 ± 11.39 | 95.05 | 85.24 ± 14.68 | 95.09 | 91.73 ± 25.11 |
| | 1.0 | 95.81 | 64.12 ± 10.25 | 95.05 | 85.24 ± 14.68 | 95.12 | 99.33 ± 22.42 |
| | 2.0 | 95.58 | 81.82 ± 12.73 | 94.92 | 90.38 ± 17.29 | 95.12 | 107.26 ± 24.30 |
| | 3.0 | 95.26 | 119.40 ± 15.65 | 95.12 | 97.44 ± 14.67 | 95.12 | 105.80 ± 21.06 |
| G2 | **0.0** | **95.01** | **69.62 ± 12.85** | **95.05** | **58.07 ± 9.83** | **95.20** | **74.86 ± 9.63** |
| | 0.5 | 95.01 | 68.04 ± 11.06 | 95.12 | 69.34 ± 13.67 | 95.16 | 79.08 ± 10.81 |
| | 1.0 | 94.97 | 69.45 ± 11.37 | 95.12 | 69.34 ± 13.67 | 95.14 | 80.04 ± 11.29 |
| | 2.0 | 95.01 | 92.31 ± 13.12 | 95.03 | 68.97 ± 13.18 | 95.09 | 79.68 ± 11.27 |
| | 3.0 | 95.09 | 117.99 ± 25.89 | 95.12 | 69.56 ± 11.77 | 95.12 | 80.06 ± 11.31 |

need: 1) additional estimation of tail-up parameters $\phi, \sigma^2$ and using historical covariance matrix weighted addition of spatial covariance and empirical covariance. Consider the node size is $n$ and the optimization methods (e.g., least square in our implementation) iteration round $N$. The former estimation takes $O(Nn^2)$, and the latter addition takes $O(n^3)$, as pseudo-inverse of matrix is

involved. However, MultiDimSPCI also needs matrix inverse for Mahalanobis distance calculation. Additionally, estimation of only two parameters can converge fast. Therefore, our method does not need significantly more time than the baseline method, consistent to the Table 5. STACI provides fast, reliable, and interpretable joint UQ over localized subgraphs across long time horizons.

Table 5: Computation Time (seconds) for Different CP Methods with Different Calibration Set Size $n$

| Calibration Set Size $n$ | Sphere | Sphere-ACI ($\gamma = 0.01$) | Square | *MultiDimSPCI* | **STACI** ($\gamma = 0$) | **STACI** ($\gamma = 0.01$) |
|---|---|---|---|---|---|---|
| 100 | 142 | 144 | 24 | 24 | 25 | 25 |
| 200 | 152 | 153 | 24 | 24 | 26 | 25 |
| 300 | 135 | 129 | 24 | 24 | 25 | 26 |
| 400 | 132 | 135 | 25 | 24 | 24 | 25 |
| 500 | 135 | 136 | 23 | 24 | 23 | 24 |

### D.8  Method Details

We provide pseudo-codes for our proposed method in 1. This applies all experiments in this paper, excluding offline setting detailed in Section D.10.

### D.9  Hyperparameter Study

The studies for hyperparameter are presented in Figure 6.

### D.10  Additional Ablation Study: Offline Experiment

In both synthetic and real-world data, MultiDimSPCI achieves the closest to our proposed method, STACI, in efficiency. Therefore, we focus our comparison on four specific variants: vanilla MultiDimSPCI($\gamma = 0$), MultiDimSPCI($\gamma = 0.01$), STACI($\gamma = 0$), and STACI($\gamma = 0.01$). In the offline setting, STACI does not update the covariance matrix estimation. To ensure a fair comparison, we similarly fix the covariance matrix for MultiDimSPCI methods at the beginning of the test phase.

The results are illustrated in Figure 7. As seen in the left figure, fixing the covariance matrix significantly improves the coverage rates of all methods, bringing them close to the desired 95% level. However, despite having the same $\gamma$, STACI consistently outperforms MultiDimSPCI in efficiency. Notably, when ACI is not applied ($\gamma = 0$), both methods tend to be overly conservative, resulting in coverage rates well above the desired 95%. Therefore, since STACI ($\gamma = 0$) achieves a higher coverage rate, MultiDimSPCI ($\gamma = 0.01$) and STACI($\gamma = 0$) exhibit similar efficiency.

In conclusion, regardless of whether the covariance matrix is fixed or not, STACI consistently surpasses MultiDimSPCI in both coverage and efficiency. Furthermore, to achieve an exact coverage rate, incorporating ACI ($\gamma = 0.01$) is recommended.

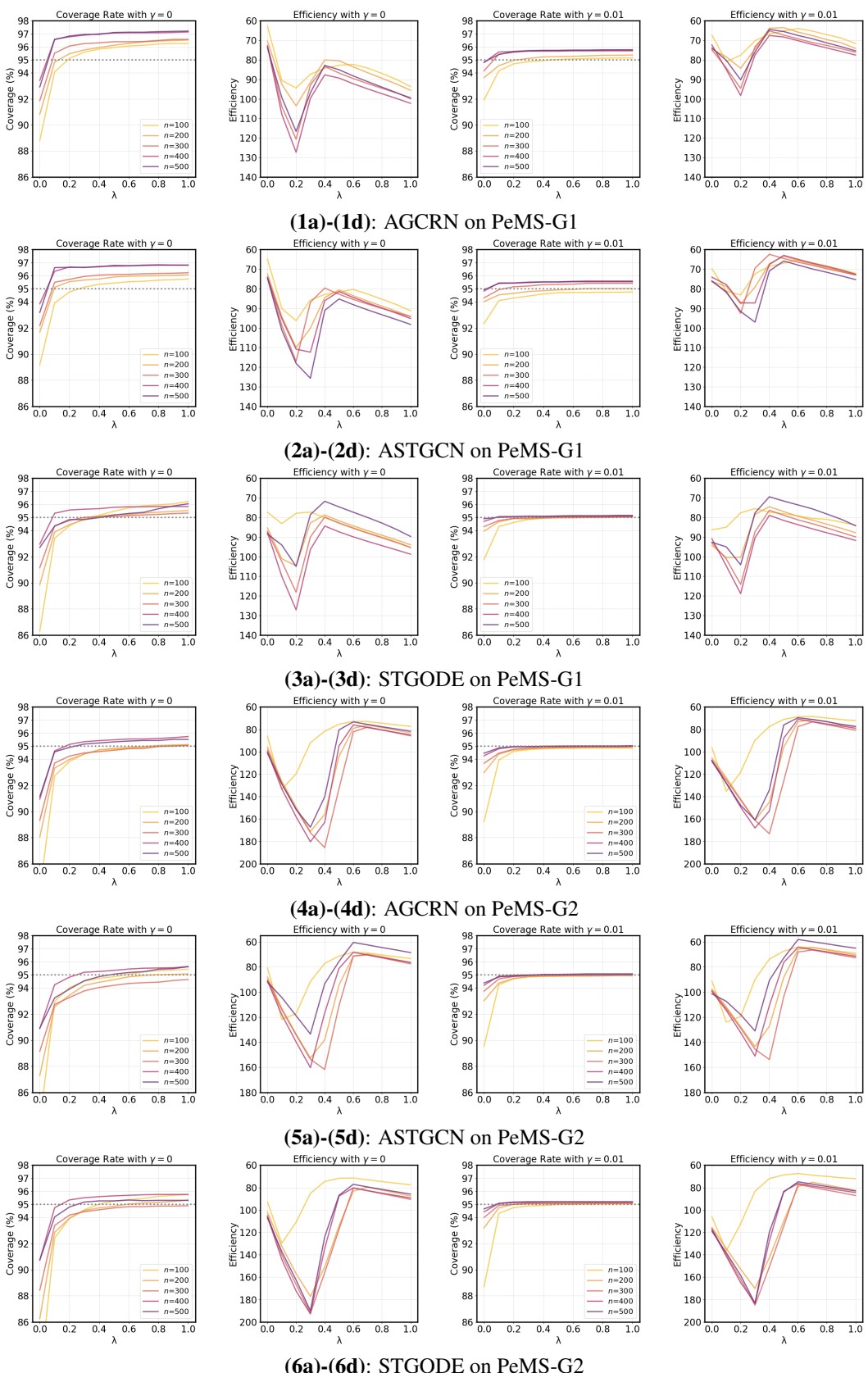

Figure 6: Comparison of Coverage and Efficiency for all PeMS experiments with different backbone GNN models and datasets. Each figure show results for different belief weight $\lambda$ and calibration set size $n$, with adaptive step size $\gamma = 0$ (a&b) and $0.01$ (c&d). Pre-determined coverage threshold of 95% is shown by horizontal gray dotted lines. 26

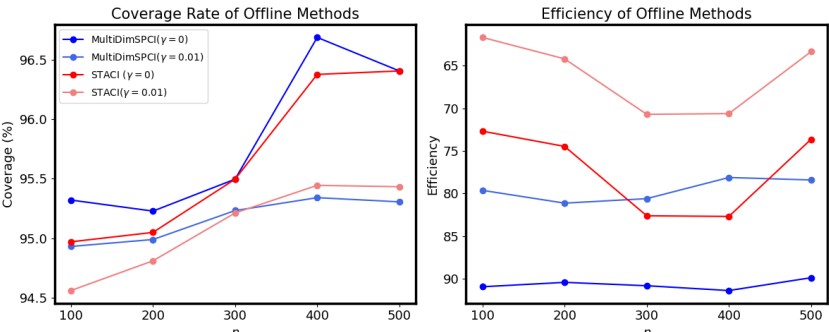

Figure 7: Coverage and efficiency of different methods with different calibration set size $n$. Multi-DimSPCI methods are in blue, and our methods are in red. Methods with $\gamma = 0$ are in darker colors; while those with adaptive coverage, $\gamma = 0.01$, are shown in shallow colors.

