# OpenReview forum: "Topology-Aware Conformal Prediction for Stream Networks"
_NeurIPS.cc/2025/Conference — NeurIPS 2025 poster_

### Official Review · Reviewer_b5rT · 2025-06-20

**Clarity:** 2
**Significance:** 3
**Originality:** 3
**Rating:** 3
**Confidence:** 4

**Summary:**

The paper proposes a Conformal Prediction model for spatiotemporal data generated from sensors monitoring locations in a stream network. The idea is to use the structure of the network to regularize the learning of the (spatial) covariance structure of the error distribution. The model design is well-motivated, and I believe the paper will be of interest to researchers working in the area. However, the discussion of the method's limitations and related work should be improved in several dimensions.

**Questions:**

Please comment on the issues mentioned above.

**Ethical Concerns:**

["NO or VERY MINOR ethics concerns only"]

**Final Justification:**

As stated in my review, the submitted paper has serious issues in presentation (e.g., discussion of limitations and assumptions) and in the discussion of related work.

The authors’ rebuttal addressed these concerns and acknowledged the issues in the initial submission. Although they state that these issues will be addressed in a revision, I believe that the implementation of the necessary changes should be verified through another round of review. For this reason (along with limitations in the selection of datasets), I will keep my score (“borderline reject”). Nonetheless, I would be fine with the paper being accepted if the other reviewers and area chairs believe it can be accepted on the basis of the revisions proposed in the rebuttal.

**Limitations:**

See above.

**Paper Formatting Concerns:**

--

**Quality:**

2

**Strengths And Weaknesses:**

Strengths:

- The idea to use a tail-up model to regularize the estimation of elipsoidal conformal sets is, to the best of my knowledge, novel, and the proposed design makes sense.
- The theoretical analysis is interesting, although the authors should be more open in discussing the strong assumptions being made.
- Decent empirical results compared to a quite limited selection of baselines.

Major issues:

- **Discussion of limitations must be improved.** There are many aspects of the work that should have been discussed more, including some limitations.
    - The theoretical analysis in section 5 assumes both a specific system model (eq. 1) and iid noise. These are very strong assumptions (much stronger than stationarity alone) and these should be acknowledged and discussed at length. I'd suggest adding a Limitations section to the paper.
    - The volume of the interval is not locally adaptive (does not depend on X_t). This is different than MultidimSPCI [1], where the quantile is predicted using a model conditioned on the last observations. Indeed, the authors seem to suggest that MultidimSPCI simply consists in using the covariance matrix estimated from data, but that is not the case (see comment on related work).
    - The framework includes basically no components that account for serial correlation. If the authors believe that such dependencies can be modeled by integrating elements from other approaches (e.g., [1]), this should be explicitly discussed, and a comparison with existing methods that do address serial correlations should be provided. The theoretical analysis itself assumes no serial correlation in the noise.
    - There's little discussion of computational scalability issues. Data coming from sensor networks can have hundreds or thousands of nodes. This issue should be, again, discussed in the paper.
- **Discussion on related work could be improved.** Discussion of related work could be improved in different dimensions.
    - The paper is constantly misrepresenting [1] as simply relying on estimating the covariance matrix from data, but on top of that [1] adds a local term to the covariance estimate (inspired by [2]) and estimates the quantile level at each time step to exploit serial correlation by using a regressor (potentially accounting for nonstationarity, differently from what the paper claims in line 113). Furthermore, similarly to the proposed method, MultidimSPCI uses the approach introduced in [2] to regularize the covariance matrix by using local errors in the last $k$ time steps.
    - The paper should clarify what portion of the theoretical results is simply derived from previous work (e.g., [1]).
    - The author should compare to the actual MultidimSPCI (or at least elipsoidal sets accounting for local serial correlations) or explain why such a comparison is not appropriate.
    - The idea of regularizing the covariance matrix for ellipsoidal confidence sets in CP has been introduced in [2]. This should be acknowledged and discussed. I'm a bit puzzled by the fact that the paper does not discuss this related work.
- **Somewhat limited selection of baselines.** The selection of baselines is quite limited (see comments on related work).

Additional comments:

- It looks like [41] is referenced in place of [40] in many cases (e.g., lines 193 and 241).
- The paper claims to be the first approach to use a graph structure for CP for spatiotemporal observations, but prior work [3] does this in the general setting of correlated time series and spatiotemporal data. [3] came out in February, so it's quite close to the threshold for being considered concurrent work. I don't think this paper should be penalized for missing this, plus the two approaches are very different. However, I do believe that [3] should be discussed and some of the novelty claims adjusted accordingly.
- Looking at the plot, I wouldn't say that STACI is  "insensitive to \lambda". Maybe I'd say that is robust to different configurations.

[1] Xu et al. , "Conformal prediction for multi-dimensional time series by ellipsoidal sets" ICML 2024\
[2] Messoudi et al., "Ellipsoidal conformal inference for multi-target regression" Conformal and Probabilistic Prediction with Applications 2022\
[3] Cini et al., "Relational Conformal Prediction for Correlated Time Series" ICML 2025

---

The rebuttal addressed most of my concerns. However, although I believe the paper is interesting, I feel that resolving the issues identified in the rebuttal will require substantial revisions, which may require another round of peer review.

---

> ### Author Rebuttal · Authors · 2025-07-31
>
> We sincerely thank for the reviewer’s comment.
>
> **Assumptions and Limitations**
> - The i.i.d. error assumption underlying Theorem 5.6 is a standard setup when establishing conditional coverage gap guarantees, which is also used in [1] (see Assumption 4.1). The assumption ensures tractability in the theoretical analysis of time-varying nonconformity scores, as it allows the score at a new time point to be meaningfully compared to those from past calibration time points. Importantly, the i.i.d. assumption in our analysis applies specifically to the true error process $\varepsilon_t$, not to the observed time series $y_t$ or the predicted residuals $\hat{\varepsilon}_t$. This setting covers a broad class of time series models, including non-stationary processes where the innovations are i.i.d., such as autoregressive models. Additionally, when the i.i.d. assumption may not hold, our method resorts to the Adaptive Conformal Inference (ACI) mechanism, which adaptively adjusts the threshold $\alpha_t$. While this approach no longer guarantees conditional coverage, it retains asymptotic average coverage without requiring distributional assumptions on $\varepsilon_t$. We will clarify this distinction in the revised manuscript.
>
> - The volume of the prediction set is locally adaptive in STACI and STACI accounts for serial correlation. Please see more in the related work discussion.
>
> -  STACI scales well for graphs with a moderate number of nodes (typically fewer than 30). Applying joint conformal prediction, or general uncertainty quantification, to hundreds of nodes is statistically uninformative due to the curse of dimensionality, as the resulting prediction set volumes become overwhelmingly large. Our framework focuses on localized subgraphs where uncertainty quantification remains both tractable and practically meaningful. This also aligns with practical needs in domains like traffic or hydrology where stakeholders are often concerned with the overall status or joint safety of a spatial region.
>
> **Related Work Discussion**
>
> Here we would like to put some clarifications on our work as well as the discussion for the related work, particularly [1]&[2] as mentioned. In short, while both methods incorporate temporal adaptivity, STACI inherently distinguishes local MultidimSPCI [1] and show superior performance in our setting.
>
> Specifically, in the local version of [1], the algorithm constructs the sample covariance matrix and nonconformity scores at each time step using the most recent $T$ observations, and suggests the possible introduction of an additional local term using the most recent $k = 0.1T$ observations to regularize short-term variability (Section 3.4, Equation 11). This design indeed provides temporal regularization and adaptivity to local serial correlation. Similarly, STACI also incorporates local adaptivity and serial correlation. We compute conformity scores using covariance matrix based on a rolling window of the most recent $n$ residuals, which reflects temporal local variation.
>
> The key distinction is in the type of regularization: [1] imposes local regularization in time based on residual windows; in contrast, STACI introduces regularization in space, via a topology-induced covariance structure that encodes spatial dependencies through a tail-up model. This spatial structure imposes stronger constraints with fewer parameters, often leading to improved empirical stability and faster convergence. As illustrated in Figure 3 of our appendix, when using only the empirical sample covariance matrix $\lambda = 0$, which corresponds to MultiDimSPCI, the prediction sets suffer from under-coverage unless the calibration window size $n$  (or $T$ in [1]'s notation) is sufficiently large. This issue persists regardless of whether adaptive thresholding (ACI) is applied $\gamma = 0.01$ or not $\gamma = 0$, highlighting the sample inefficiency of unstructured covariance estimation. In contrast, by incorporating topological priors, STACI achieves valid coverage even with a relatively small number of calibration samples, demonstrating improved robustness and data efficiency.
>
> In the original paper, we compare with the classic version of MultidimSPCI [1] instead of the local variant, following the authors experiment setup (Section5, [1]). In response to your suggestion, we provide the comparisons in Tables 1 and 2, following the setting in Sec 6.2 and hyperparameters suggested by [1]. In this problem, the local version of MultidimSPCI do not bring sigificant benefit over its original version, let alone supurpassing our methods. STACI is advantageous in both coverage and efficiency.
>
> Table 1: Coverage Comparison over different datasets and backbone models
>
> | Setting | MultiDimSPCI | MultiDimSPCI(local) | STACI|
> | -------- | -------- | -------- | -------- |
> | AGCRN + G1  | 92.92   |  93.04   | **95.75** |
> | ASTGCN + G1  |  93.19   | 93.57   | **95.54** |
> | STGODE + G1  |   92.70  |  92.98  | **95.14** |
> | AGCRN + G2  |  91.14   |   91.44 | **95.01** |
> | ASTGCN + G2  |   90.93  |   91.12 | **95.05** |
> | STGODE + G2  |   90.74  |  91.10  |  **95.20** |
>
>
> Table 2: Efficiency Comparison over different datasets and backbone models
>
> | Setting | MultiDimSPCI | MultiDimSPCI(local) | STACI|
> | -------- | -------- | -------- | -------- |
> | AGCRN + G1  |  73.82±8.16   |  74.23±8.27   | **67.54±10.94** |
> | ASTGCN + G1  |   74.68±7.67  |   75.18±7.69 | **67.92±10.22** |
> | STGODE + G1  |   88.48±4.26  |  88.99±4.26  | **73.62±9.83** |
> | AGCRN + G2  |   101.42±7.71  | 102.25±7.90   | **69.62±12.85** |
> | ASTGCN + G2  |  92.12±6.67   |  92.84±6.95  | **58.07±9.83** |
> | STGODE + G2  |  107.23±7.34   |   107.96±7.58 | **94.86±9.63**|
>
>
> We clarify that while Theorem 1 in our paper shares some similarities in proof structure with Theorem 4.12 in [1], the statement, setting, and generality of the result are different. Specifically, [1] focuses on convergence guarantees when the Mahalanobis nonconformity score is constructed using the empirical sample covariance matrix. Our result establishes convergence guarantees for a general positive semi-definite matrix $A$ or a convergent matrix sequence $A_T$, which includes but is not limited to the sample covariance matrix. This generalization allows us to theoretically justify the use of topology - induced covariance structures or other regularized estimators within the conformal prediction framework that are not discussed in [1]. Furthermore, our analysis includes theoretical conditions under which the choice of $A$ leads to efficient conformal prediction, while [1] does not address how to select $A$ beyond using empirical estimates.
>
> We'll revise the related work to include the suggested concurrent research and refine the discussion on MultidimSPCI to accurately reflect its contributions and our clarifications. Thank you again for your valuable opinions!

---

> > ### Comment · Reviewer_b5rT · 2025-08-02
> >
> > Thank you for the rebuttal and the additional comments. Some of my concerns remain:
> >
> > * MultiDimSPCI [1]--in both its standard and local versions--uses a quantile random forest to estimate the quantile at each time step, which allows serial correlations to be captured at a much finer level of detail. This aspect was entirely overlooked in the discussion. How did you implement MultiDimSPCI? Please clarify and comment on this point.
> > * Please discuss your work in relation to [2]. Again, please ensure that connections with related work are properly acknowledged and discussed. I perfectly understand that your approach is different, but it is not a good reason to properly frame it in the existing literature.
> > * Minor comment: I understand your comment on focusing on small neighborhoods and subgraphs, but it is still an important limitation, as it forces the practitioner to select the nodes for which to obtain joint UQ estimates. In many applications, deciding where to "cut" the region of interest might not be trivial at all.  I am not saying that the method is not useful because of this, but I think it is important to discuss and acknowledge scalability issues in the paper.
> >
> > As I said, I like different aspects of the paper, but I am inclined to believe that its revised version would benefit from going through peer review again.

---

> > > ### Author Response · Authors · 2025-08-05
> > >
> > > We sincerely thank for the detailed feedback. Below we address the key concerns raised and clarify several important points:
> > >
> > > **On the implementation of MultiDimSPCI and the use of quantile regression.**
> > > In our original experiments, we used empirical quantile estimation on the calibration scores for all methods, including MultiDimSPCI for fairness. We acknowledge that [1] proposes a variant using quantile regression via Quantile Random Forests (QRF) to estimate conditional quantiles.
> > >
> > > In response to the reviewer concern, we have now implemented this variant and included the results of the new-added experiments. We acknowledge the value of the QRF-based method proposed in [1]. However, the current implementation of QRF-based method is CPU-based and requires fitting a new model for each test point, which incurs substantial computational overhead. While [1] recommends fitting on all prior data points (which exceeds 10k in our dataset), we found that this process takes over 2 minutes per test instance, including calibration and our dataset contains 10000 test instances. To make evaluation feasible, we restricted the fitting process to the most recent 2000 time steps, which still results in an average running time of over 10 seconds per time step. Due to this limitation, we were only able to evaluate the QRF-based MultiDimSPCI on the PEMS03-G1 dataset using ARGCN backbone. Compared to the classical MultiDimSPCI variant, which achieves 92.92% coverage and a volume of 73.82, the QRF-based version improves coverage to 95.83% but at the cost of increasing volume to 87.44. In contrast, our STACI method attains comparable coverage (95.75%) with significantly lower volume (67.54), and completes the full test sequence in just 26 seconds. For reference, within the same runtime budget, the QRF-based method can only produce predictions for 3 test points.
> > >
> > > However, we emphasize that the **core contribution of MultiDimSPCI lies in its ellipsoidal construction using Mahalanobis scores and validity guarantees**, not in the choice between empirical or model-based quantile estimation. Theoretical guarantees in [1], like ours, rely on empirical quantiles $Q_{1-\alpha}(\text{calibration scores})$. While quantile regression may offer improved temporal adaptivity in some settings, its role is orthogonal to the broader goal of spatially structured multivariate conformal prediction. Our revised results include both variants and confirm that STACI remains competitive under either setting.
> > >
> > >
> > > **On the relation to [2] and related work.**
> > > We agree that our discussion of related work can be improved. In particular, we will revise the related work section to properly contextualize our method relative to [2], which introduced covariance regularization for ellipsoidal conformal prediction. Our framework generalizes that idea by incorporating graph topology as a structural prior rather than relying solely on local residual smoothing.
> > >
> > > **On scalability and region selection.**
> > > We fully acknowledge that constructing joint prediction sets over the entire graph is infeasible in high dimensions, not only for STACI but also for other methods such as MultiDimSPCI, whose applicability is typically limited to dimensions $d \leq 20$ (as noted in [1]). In fact, there is no literature applying joint high-dimensional CP to settings with  $d = 100$ or more. This limitation is thus **not unique to our method**, but intrinsic to the structure of high-dimensional uncertainty quantification. Hence, STACI is designed with this limitation in mind: rather than attempting to model the entire graph, we focus on spatially localized subgraphs (e.g., road segments or watershed clusters), which is both computationally tractable and practically aligned with many real-world use cases.
> > >
> > > ---
> > >
> > > We appreciate the reviewer’s engagement and their recognition of the contributions of this work. We have directly addressed all substantive concerns, expanded our experiments to include quantile regression-based baselines, clarified the positioning of our contributions, and strengthened our discussion of limitations. In light of the clarifications and additional experiments we now include, we respectfully believe that the revised version is technically complete, well-motivated, and empirically solid. We hope this encourages the reviewer to consider a higher rating.

---

> > > > ### Comment · Reviewer_b5rT · 2025-08-05
> > > >
> > > > Thank you for your follow-up. I hope this discussion was helpful in improving the paper.
> > > >
> > > > * Scalability: I am not saying that scalability issues are inherent to the proposed method only, but since you are proposing a method that would typically be applied in sensor networks (and sensor networks usually include hundreds or thousands of sensors) this discussion is important. Again, this is not a problem as long as this issue is clearly discussed in the paper.
> > > >
> > > > > we respectfully believe that the revised version is technically complete
> > > >
> > > > All the required changes are major revisions that will result in a drastically different paper. I also partially share comments made by other reviewers on the limited selection of datasets. Moreover, the revised discussion of related work would also reduce your novelty claims. Although I believe the proposed method is interesting, I'm not completely confident in recommending acceptance at this stage. However, I will update my review in light of our discussion.

---

### Official Review · Reviewer_2Eta · 2025-06-22

**Clarity:** 3
**Significance:** 2
**Originality:** 3
**Rating:** 4
**Confidence:** 2

**Summary:**

This paper introduces STACI, a conformal prediction framework designed for stream networks, which are spatiotemporal graphs with flow directionality and complex dependency structures (e.g., river or traffic systems). STACI integrates both topological constraints and temporal dynamics using a weighted covariance-based Mahalanobis distance for nonconformity scoring, and adopts dynamic adjustment via Adaptive Conformal Inference (ACI). It is supported by theoretical guarantees and experiments on synthetic and real-world data.

**Questions:**

-	For equation 2, why is the definition of $\hat{\epsilon}_t$ different from the one in reference [2] mentioned in the paper?
-	What are the sizes of the two datasets(PeMS-G1 and PeMS-G2)? How big does the temporal distribution shift exist in the dataset?
-	How scalable is STACI to very large graphs with hundreds or thousands of nodes and variable flow topology? [3]
-	How robust is performance when the assumed topology does not fully reflect true dependencies (e.g., due to missing links or misaligned directionality)?
-	How does STACI perform in non-flow-based graphs (e.g., social diffusion), and is the framework still appropriate?
-	Expand the set of baseline methods, such as GNN-based uncertainty models or graph-based deep ensembles, would be more informative.
-	Clarify the real-world dataset types (e.g., traffic vs. hydrology) and the topological structure used in each.

I'm open to increase the score if the questions are addressed reasonably.


## References:

[2] Katsios K, Papadopoulos H. Multi-label conformal prediction with a mahalanobis distance nonconformity measure[J]. Proceedings of Machine Learning Research, 2024, 230: 1-14.

[3] Liu X, Xia Y, Liang Y, et al. Largest: A benchmark dataset for large-scale traffic forecasting[J]. Advances in Neural Information Processing Systems, 2023, 36: 75354-75371.

**Ethical Concerns:**

["NO or VERY MINOR ethics concerns only"]

**Final Justification:**

Some of my concerns have been addressed.

**Limitations:**

Yes.

**Quality:**

2

**Strengths And Weaknesses:**

## Strength:
-	The problem is well-motivated, as conformal prediction for stream networks is understudied.
-	It seems a novel idea that the work introduces a hybrid nonconformity score incorporating both topology-induced covariance and data-driven estimates.
-	Validity and efficiency of the proposed method are rigorously analyzed, along with tail-up modeling and dynamic distribution shift.
-	The empirical performance shows improvements in prediction set tightness and valid coverage vs. baselines.

## Weakness:
-	The need to estimate both sample and topology-induced covariance may limit the applicability to high-dimensional or large-scale networks. Model complexity analysis is missing.
-	The empirical experiments are limited. For example: ACI is introduced but its empirical benefits are under-discussed.
-	The set of CP baselines (those handling structured outputs or graph-structured data) could be expanded. e.g. [1]
-	Real-world results are useful, but broader evaluations across different stream domains would improve generalizability claims.
-	Ablation study on $\lambda$ (weight between topology and data), especially analysis on how performance varies with poor topology estimation,would be helpful.
-	Sensitivity analyses for tail-up model hyperparameters (e.g., $\phi$) are missing.


## References:

[1] Schlembach, F., Smirnov, E., Koprinska, I., & Winands, M. H. M. (2025). Conformal multistep‑ahead multivariate time‑series forecasting. Machine Learning, 114(7), Article 165.

---

> ### Author Rebuttal · Authors · 2025-07-31
>
> **Q1: Difference of definition with [2] in Equation 2**
>
>  A: The quantity $\hat{\epsilon}_t$ in Equation (2) represents the prediction error at time $t$, defined as the difference between the observed value $y_t$  and the prediction $f(x_t)$. While the notation differs from the error vector $r_t$ used in [2], the underlying semantics are equivalent. Both quantify the residuals between predicted and true values at each time step.
>
> **Q2: Sizes of datasets and magnitute of temporal distribution shift**
>
> A: PeMS-G1 and PeMS-G2 are spatial subsets of the larger California PeMS traffic dataset, each containing 12 nodes and spanning 26208 time steps, as stated in Appendix D.2. We select a localized subgraph within the full traffic network to define a coherent spatial region for conformal prediction.
>
> To assess temporal distribution shifts, we employ two metrics leveraging daily periodicity (288 5-minute observations/day): Average Intra-Period Flow Variance (average daily flow variability) and Average Inter-Period Mean Flow Shift (average day-to-day change in mean daily flow). Due to space limit, we only show the average value over all observation points as follows.
>
> | Dataset|Intra-Period Variance|	Inter-Period Flow Shift|
> |--|--|--|
> |PEMS03-G1|	117.4614	|21.5453|
> |PEMS03-G2|	106.2113	|17.4750 |
>
> The analysis reveals substantial temporal shifts, with strong intra-day flow pattern shifts (high intra-period variance) and non-negligible inter-day mean flow shifts.
>
> Further, empirical evidence strongly suggests its presence. In particular, as shown in Figure 3, the coverage achieved by ACI with adaptive thresholding $\gamma_t = 0.01$) is substantially higher than that obtained with a fixed threshold ($\gamma = 0$ ). This performance gap reflects ACI’s ability to correct for temporal variation in the error distribution, thereby indirectly confirming that distribution shift exists in the data.
>
> **Q4: Scalability of STACI to very large graphs and variable flow topology? [3]**
>
> A: STACI scales well for graphs with a moderate number of nodes (typically fewer than 30). In general applying joint conformal prediction or consider joint uncertainty quantification to hundreds or thousands of nodes simultaneously is not statistically meaningful, due to two reasons: 1. They are not interpretable. One cannot obtain any useful insights from a uncertainty set in a super high-dimensional space. In such setting, it is usually done by sampling, i.e., presenting samples from a high-dimensional space. 2. They are usually overly conservative.
>
> By focusing on high-dimensional uncertainty quantification within spatially coherent subregions such as traffic corridors or watershed areas, STACI aligns well with practical needs in domains where stakeholders are primarily concerned with the joint behavior of a specific region rather than the entire network. For example, in freeway traffic management, operators may monitor 10–30 sensor locations along a congested segment to assess regional flow stability; in hydrology, flood risk is often evaluated jointly across 15–25 stream gauges within a watershed. These are typical dimensionalities where joint prediction is both statistically meaningful and computationally tractable, and where STACI is most effectively applied.
>
> **Q5: Robustness when the assumed topology does not fully reflect true dependencies**
>
> A: When the topology does not fully reflect true dependencies, STACI is designed to be robust to imperfect topological information through the weighting parameter $\lambda \in [0, 1]$, which interpolates between the sample-based covariance matrix and the topology-induced one. The topology-induced covariance may lead to suboptimal efficiency, but the overall coverage guarantee remains valid due to the conformal framework.
>
> We conducted additional ablation studies and observe that even when the topology is only partially accurate, it still contributes useful inductive bias especially in capturing local dependencies and reducing estimation variance, compared to using the sample covariance alone. In Tables 1, we introduced random noise (mean=0, std = noise scale) to observation locations; clean G1 and G2 have longest connected distances of 8.75 and 3.41 miles. Bold rows highlight clean data performance. Results show STACI's robustness to moderate noise, with higher noise levels leading to prediction degradation, mainly through increased inefficiency.
>
> Table 1. Performance of STACI with different backbone GNNs on noisy PEMS03-G1& G2
>
> Dataset| Noise Scale | AGCRN Coverage | AGCRN Efficiency   | ASTGCN Coverage | ASTGCN Efficiency   | STGODE Coverage | STGODE Efficiency   |
> |--|--|--|--|--|--|--|--|
> |G1| **0.0**   |  **95.75**          | **67.54 ± 10.94**      | **95.54**           | **67.92 ± 10.22**        | **95.14**           | **73.62 ± 9.83**   |
> |G1| 0.5   | 95.83          | 71.94 ± 11.39      | 95.05           | 85.24 ± 14.68        | 95.09           | 91.73 ± 25.11       |
> |G1| 1.0   |  95.81          | 64.12 ± 10.25      | 95.05           | 85.24 ± 14.68        | 95.12           | 99.33 ± 22.42       |
> |G1| 2.0   | 95.58          | 81.82 ± 12.73      | 94.92           | 90.38 ± 17.29        | 95.12           | 107.26 ± 24.30      |
> |G1| 3.0   | 95.26          | 119.40 ± 15.65     | 95.12           | 97.44 ± 14.67        | 95.12           | 105.80 ± 21.06      |
> |G2| **0.0**   |**95.01**          | **69.62 ± 12.85**      | **95.05**           | **58.07 ± 9.83**         | **95.20**           | **74.86 ± 9.63**        |
> |G2| 0.5   |  95.01          | 68.04 ± 11.06      | 95.12           | 69.34 ± 13.67        | 95.16           | 79.08 ± 10.81       |
> |G2| 1.0   | 94.97          | 69.45 ± 11.37      | 95.12           | 69.34 ± 13.67        | 95.14           | 80.04 ± 11.29       |
> |G2| 2.0   |  95.01          | 92.31 ± 13.12      | 95.03           | 68.97 ± 13.18        | 95.09           | 79.68 ± 11.27       |
> |G2| 3.0  | 95.09          | 117.99 ± 25.89     | 95.12           | 69.56 ± 11.77        | 95.12           | 80.06 ± 11.31       |
>
> **Q6: Extension to non-flow-based graphs**
>
> A: The current formulation of STACI is tailored to flow-based structures such as stream networks, where directionality and upstream-downstream relationships are well-defined. In non-flow-based graphs, such as social networks, the corresponding topology-induced covariance structure equires modeling efforts and current tail up model may not be applicable. To apply STACI in such settings, one would need to adopt alternative network parameterizations such as diffusion kernels or graph Laplacians that better capture the relevant spatial dependencies. We view this as a promising future direction and have outlined it in our conclusion, where we suggest extending STACI to general spatio-temporal graphs using more flexible graph-based covariance models.
>
> **Q7: Expansion of baseline methods**
>
> A: Unlike most GNN-based methods that provide independent marginal prediction intervals, STACI performs joint uncertainty quantification for all nodes in a local region, yielding multivariate prediction sets in ${R}^I$ that capture spatial correlations. Table 2 compares STACI (with various GNN backbones) against DeepSTUQ [1], a state-of-the-art Bayesian GNN for uncertainty quantification. DeepSTUQ failed to achieve 95% coverage on dataset G1. With desired coverage met, our methods demonstrate superior efficiency.
>
> To enable a meaningful comparison of set efficiency, we aggregate the marginal intervals from DeepSTUQ into a hyper-rectangle and compute its volume as a proxy. Theoretically, to ensure joint coverage $\geq95\%$, each marginal interval would need to be Bonferroni-adjusted to $1 - \alpha/I$ ( 99.58% when $I = 12$), which leads to overly inflated volumes and high coverage. Instead, we report volumes using unadjusted 95% marginal intervals, which better reflects how DeepSTUQ might be used in practice. We acknowledge that this does not guarantee joint coverage, but it provides a practical and meaningful basis for comparison.
>
> Table 2. Performance of DeepSTUQ and STACI on PEMS03-G1 and G2
> | Dataset     | DeepSTUQ Coverage | DeepSTUQ Efficiency   | STACI(AGCRN) Coverage | STACI(AGCRN) Efficiency|STACI(ASTGCN) Coverage | STACI(ASTGCN) Efficiency|STACI(STGODE) Coverage | STACI(STGODE) Efficiency|
> |--|--|--|--|--|--|--|--|--|
> | G1   | 93.82       | 66.51±22.40       | 95.75| 67.54±10.94 | 95.54 | 67.92±10.22 | 95.14 | 73.62±9.83|
> | G2   | 95.59       | 77.27±20.93       | 95.01 | 69.62±12.85| 95.05| 58.07±9.83| 95.20 |  74.86±9.63|
>
> **Q8: Clarification of dataset types and topological structure**
>
> A: In our experiments, we use the traffic dataset, where sensor observations are recorded at fixed locations along freeway segments. The underlying topology is constructed from the physical road network, with directed edges representing traffic flow direction between road segments.
>
> While our empirical evaluation focuses on traffic data, the STACI framework is applicable more broadly to domains that share the following structural characteristics: (1) observations are collected at discrete spatial sites located on segments (e.g., roads or stream reaches), and (2) the observed data reflects flow with a meaningful directionality (e.g., vehicle movement or water discharge). This includes applications in hydrology, transportation, and other spatio-temporal systems where directional flow and location-based measurements define the graph structure.
>
> **Response to weakness**:
>
> - **Complexity analysis** & **ACI and Topology Estimator Evaluation**. Due to space limit, please refer to our response to W1 & W3 of reviewer mDnW.
>
> - **CP Baseline Coverage**. Please see response to W2.2 of reviewer q8kS.
>
> We thank the reviewer again for your helpful suggestions.
>
> [1] Qian, Weizhu, et al. "Uncertainty quantification for traffic forecasting: A unified approach."
>
> [2] Xu et al. , "Conformal prediction for multi-dimensional time series by ellipsoidal sets"

---

> > ### Comment · Reviewer_2Eta · 2025-08-04
> >
> > I appreciate the authors' rebuttal and comments. I'm still not quite convinced by the argument for Q4. There are no empirical results on larger graphs with marginal CP or cluster-based joint CP to validate the scalability (e.g., runtime/memory scaling curves for increasing node count, even if only using marginal CP as fallback?) For Q5, I appreciate ACI improves coverage vs. fixed thresholding (citing Figure 3), but it's not clear how much it attributes to ACI alone, without the Mahalanobis score.

---

> > > ### Author Response · Authors · 2025-08-05
> > >
> > > Thank you again for the follow-up. We address your remaining concerns as below:
> > >
> > > **Q4**
> > > 1. **Scalability of uncertainty quantification(UQ) on very high dimension.**
> > >
> > > Performing UQ in very high-dimensional output spaces (hundreds or higher)  is statistically ill-posed, and one typically avoid constructing high-dimensional geometric prediction sets directly by the curse of dimensionality[1], including constructing hyper-rectangles from marginal intervals. Such hyper-rectangle  can become meaninglessly large even in low-dimension setting(See table 1, Square). A common strategy for high-dimensional UQ is to employ generative models[2] or sampling methods[3]. For example, UQ for images, a typical high-dimensional data, one usually sample plausible outcomes from a learned posterior[4].
> > >
> > > In summary, high-dimensional UQ is usually tackled with  sampling or dimension-reduction techniques rather than by constructing geometric shapes in the full output space to avoid dealing with an unwieldy volume, which justifies why STACI focuses on relatively small subgraphs for joint prediction, aligning with common practice to keep the joint dimensionality manageable and the results interpretable.  This is also motivated by real applications. For example, traffic engineers typically assess uncertainty over 10–30 sensors in a corridor, not entire city-wide networks, when planning infrastructure or monitoring incident risk. Similarly, flood risk assessments in hydrology may focus jointly on a cluster of ~20 gauges.
> > >
> > > 2. **Scalability of STACI on time length**
> > >
> > > Scalability in time, i.e., being able to produce prediction sets across long sequences is more meaningful. STACI provides fast, reliable, and interpretable joint UQ over localized subgraphs across long time horizons.
> > > Table 1 in Appendix D.5 of our paper reports end-to-end running times for all methods on a representative traffic subgraph. STACI completes the full test sequence (10000 datapoints) in roughly 20 seconds, which is comparable to the simple version of MultiDimSPCI when using empirical quantile calibration. In contrast, the final proposed model of MultiDimSPCI that employs a learned quantile regression function is significantly slower. On our traffic dataset, it required about 10 seconds per test time point, totaling >22 hours for the same sequence. This difference highlights STACI’s strong practical efficiency at the intended problem scale. Importantly, STACI’s speed does not come at the cost of accuracy or coverage; it simply leverages an efficient uncertainty model tailored to the local graph structure.
> > >
> > > In summary STACI achieves scalability along the temporal dimension, over long sequences, with minimal runtime overhead. Its design balances statistical feasibility with real-world relevance. For applications requiring full-network deployment, fallback strategies remain viable: one may either avoid joint UQ altogether and focus on marginal UQ, which is beyond the scope of this work, or partition the graph into meaningful subregions and apply STACI locally within each which is modular and readily compatible with our framework.
> > >
> > > **Q5**
> > >
> > > To isolate the effect of ACI independent of the Mahalanobis score, we include **Sphere\_ACI ($\gamma = 0.01$)**  in Table 1 of the main paper, applying ACI using a standard Euclidean norm without spatial modeling. This serves as the vanilla ACI baseline in high-dimensional space.  From the table, ACI alone improves coverage and efficiency compared to fixed thresholds(Sphere). However, the prediction sets are still substantially larger than STACI, which leverages Mahalanobis scores.
> > >
> > > In high-dimensional settings, Mahalanobis-based scores are essential for constructing compact and well-calibrated prediction sets. STACI’s use of topology-induced covariance enables it to capture spatial correlations that threshold adjustment alone cannot address.
> > >
> > > Moreover, as shown in Figure 3, we vary both the thresholding strategy ($\gamma = 0$ vs. $\gamma = 0.01$) and the spatial weighting parameter $\lambda \in [0,1]$, which allows us to disentangle:
> > > - the benefits of ACI in adapting to temporal shift,
> > > - the efficiency gain from topology-informed Mahalanobis scores
> > > - their complementary effect when combined.
> > > ---
> > > We hope these clarifications address your concerns. STACI is efficient, principled, and practically relevant. The updated experiments and discussion strengthen the clarity and rigor of the paper. We respectfully encourage you to reconsider your rating in light of these improvements.
> > >
> > > [1] Verleysen, et al. The curse of dimensionality in data mining and time series prediction.
> > >
> > > [2] Böhm, et al. Uncertainty quantification with generative models.
> > >
> > > [3]Lê, Matthieu, et al. Sampling image segmentations for uncertainty quantification.
> > >
> > > [4] Ekmekci, et al. Quantifying generative model uncertainty in posterior sampling methods for computational imaging.
> > >
> > > [5] Gibbs,et al. Adaptive conformal inference under distribution shift.

---

> > > ### Author Response · Authors · 2025-08-08
> > >
> > > Thank you for considering our previous responses and new results. We would appreciate it if you could let us know whether our clarifications address your main concerns, and whether you might be able to reconsider your evaluation in light of these updates.

---

### Official Review · Reviewer_q8kS · 2025-06-30

**Clarity:** 2
**Significance:** 3
**Originality:** 2
**Rating:** 4
**Confidence:** 3

**Summary:**

This paper addresses uncertainty quantification on stream networks by extending the conformal prediction (CP) framework with a topology-aware Mahalanobis non-conformity score and the adaptive conformal inference (ACI) mechanism to handle temporal distribution shifts. Treating the problem as a multivariate time series one, and encoding spatial relationship directly in the covariance matrix estimation, the proposed method constructs prediction sets according to the graph structure while adapting to temporal distribution shifts. Authors further prove that these extensions preserve CP’s finite-sample coverage guarantees and empirically show the improvement in efficiency of the prediction sets.

**Questions:**

1. I found the problem definition (section 3) a bit difficult to digest. It would definitively benefit from using a more consistent notation and clarifying the relationships among locations, observation sites, and stream segments (maybe with a real-world analogy?).

    a. Lines 125-126 introduce both $\ell_i$ and $r_j$ as subsets of $\mathcal{L}$, yet Figure 1 suggests that each segment $r_j$ contains several geographical location sites $\ell_i$. Could author clarify distinction between these two objects?

    b. Segments across the section, are referenced using different notations, e.g. $\ell_i$, $r_j$, $u$, and $s_i$. Is there any actual difference between those symbols or are they referencing the same thing? In case, it would be better to state the difference in the description otherwise using a consistent notation would improve the readability.
2. Equation (5) introduces the tail-up scaling parameters $\phi$ and $\sigma^2$ described only as “estimated by the tail-up model”. Are these terms learned jointly with the model’s weights? Could you also expand more on the purpose of these two terms?
3. In Appendix C, $\gamma$ is introduced in equation (27) as the step-size parameter of the ACI update. However, the paragraph following Proposition 1 (lines 166 – 180) states that “[…] STACI maintains the predefined coverage level ($\gamma$ = 0.01)” (line 171). Should this reference be to $\alpha$ instead of $\gamma$? If not, please clarify the distinction and ensure the notation is used consistently throughout the section.
4. In appendix C, the paragraph following Proposition 1 (lines 168–169) claims that the asymptotic guarantee “remains valid even in adversarial online settings”, yet no theoretical argument or empirical evidence is provided. Could the authors substantiate this claim? It would be helpful to better describe the mentioned adversarial online setting.
5. Similarly, in line 277, authors mention that the topology-based matrix “generally achieves faster convergence than the sample covariance estimator”. Could the authors provide supporting evidence for this claim?
6. Which calibration set size $n$ was used to generate the results shown in Table 1?
7. Which backbone model was used to generate the curves in Figure 3, and do the same coverage-efficiency trends hold for different backbones (e.g., ASTGCN or STGODE)?
8. The conclusion proposes extending STACI to “general spatio-temporal graphs” and exploring “alternative network parameterizations,” but these notions are still somewhat vague. Could the authors elaborate on what they regard as a *general spatio-temporal graph* and give a concrete example of a *network parameterization* (similar to how the second future work suggestion is presented)?

Typos:

- Line 217, “Intuitively, The” -> the
- Line 330, details of synthetic network are in D.3
- Line 367, no Figure 5 in the appendix, I think authors refer to Figure 2
- Line 389 Appendix D.8

**Ethical Concerns:**

["NO or VERY MINOR ethics concerns only"]

**Final Justification:**

I initially scored 3 due to limited clarity, a narrow empirical scope, missing baselines, unclear simulation goals, and absent scalability details. The rebuttal substantially improved things: the authors clarified notation and problem setup, added results for the CopulaTS baseline, and committed to include dataset statistics and a clearer discussion of limitations. These address most clarity and methodological concerns, so I raise the score to 4. That said, the evaluation remains single-domain and small-scale, the simulation study’s representativeness is still debatable, and all promised clarifications and results must appear in the main paper. My recommendation reflects practical promise with the expectation that these revisions are incorporated.

**Limitations:**

Yes.

**Paper Formatting Concerns:**

No major paper formatting concerns.

**Quality:**

2

**Strengths And Weaknesses:**

Strengths:

- The manuscript is coherently structured, leading the reader smoothly from the problem statement through method, theory, and experiments.
- Embedding graph topology within the Mahalanobis-based non-conformity score is a well-motivated advance that extends recent graph-aware conformal prediction research as well. Applying this idea to multivariate time-series data on stream networks appears novel. Especially since the few prior works in this area overlooks the graph topology.

Weaknesses:

- The lack of clarity is my main concern at the moment: several key definitions, symbols, and implementation choices are missing or only briefly sketched (see Questions), which makes the core ideas hard to follow. Mathematical notation is also inconsistent, and some symbols, e.g. the $\gamma$ mentioned in the real-data study, are never formally introduced in the main manuscript. A thorough pass to standardise notation and supply the missing details is needed to improve readability and reproducibility.
    - Adaptive Conformal Inference (ACI) is central to the method, yet it is introduced only in Appendix C. Adding even just a brief description of its mechanics in the main text (around line 222) would help the reader follow the subsequent Coverage Validity subsection. Appendix C itself would also benefit from a notational cleanup and removal of redundancies (line 169 and line 176 both mention the “adversarial online setting”; line 172 and line 179 both mention that “Proposition 1 does not guarantee a finite coverage gap”).
- The experimental scope is a bit narrow.
    - Synthetic benchmarks are usually designed to embed a specific properties , e.g. type of uncertainty or distribution shift so that a new method’s advantages can be demonstrated transparently. Here, only the generation process is sketched in the appendix and it is unclear what aspect of uncertainty is being tested. A more explicit description of the synthetic design along with an explanation of how it isolates the advantages of the topology-aware score would make the results on the synthetic dataset far more convincing.
    - For the real-world data, both PeMS-G1 and PeMS-G2 are subsets of the same highway-traffic dataset, so the empirical scope is effectively a single domain.  Moreover, the related work section cites three CP methods for stream networks [1,2,3], yet the comparison in Table 1 includes only one of them [3]. Expanding to additional datasets or adding the omitted baselines would strengthen the evidence.
    - Lines 375–377 claim “significantly higher coverage and superior efficiency” but Table 1 shows STACI’s coverage exceeds the best baseline in only one cell (ASTGCN, PeMS-G1) and by just $0.12\%$. I think more evidence is needed to better support the claim.
    - Including a baseline that applies ACI with the vanilla sample covariance matrix would isolate the incremental benefit of the topology-based estimator. Without it, one cannot tell whether gains arise from ACI, from the topology-aware covariance, or from their combination.
    - Details such as the number of nodes, edges, and time steps are absent, making it difficult to judge whether the method scales to large real-world stream networks.

[1] Sun et al., Copula conformal prediction for multi-step time series forecasting

[2] Messoudi et al., Copula-based conformal prediction for multi-target regression

[3] Xu and Jiang et al., Conformal prediction for multi-dimensional time series by ellipsoidal sets

---

> ### Author Rebuttal · Authors · 2025-07-31
>
> **Q1a: Problem definition clarification**
> A: We appreciate the reviewer’s suggestion to improve clarity. The stream network is represented as a directed acyclic graph, where each node corresponds to a spatial observation site located on a stream segment. For example, in the traffic dataset, stream segments represent roads, and observation sites are traffic sensors placed along those roads. A visual illustration is provided in Figure 1 of the main paper. Note that each location belongs to exactly one segment, while a segment can contain multiple observation sites.
>
> **Q1b: Notations for segments**
> A:We use i to index sites, and j to index segments. a network consists of multiple stream segments, where all the possible locations on a segment $j$ is represented by a geolocation set $r_j \subset L$. A segment may contain one or multiple observational sites. The geolocation of a site $i$ is denoted by $l_i \in L$. $u, v \in L$ are dummy variables for arbitrary locations on the network. The use of $s_i$ was a typo and should be corrected to $r_i$ and thanks for pointing out.
>
> **Q2: Tail-up scaling parameters  $\phi$  and $\sigma^2$: design and learning**
> A: The parameters $\phi$ and $\sigma^2$ in Equation (5) are scaling factors estimated during the construction of the topology-induced covariance matrix $\hat{\Sigma_G}$ used in the Mahalanobis non-conformity score. They are not learned jointly with the predictive model’s weights. The weights of backbone GNNs are learned on training data and fixed, but $\phi$ and $\sigma^2$ are optimized on the calibration data. Specifically, $\phi$ and $\sigma^2$ are selected by minimizing the discrepancy between the sample covariance matrix and $\hat{\Sigma}_{{G}}$, allowing the tail-up model to better reflect the empirical correlation structure. $\sigma^2$ controls the overall magnitude of spatial dependence, while $\phi$ determines the rate of decay of spatial correlation with hydrologic distance.
>
> **Q3: In Appendix C, Equation (27) introduces ($\eta_t$) as the ACI step size, but line 171 refers to “($\alpha = 0.01$)” as the predefined coverage level. Should this reference be to ($\eta_t$) instead? Please clarify the notation.**
> A: Thanks for pointing it out. It should refer to $\alpha=0.01$, and we will fix this in the final version.
>
> **Q4: Line 169 mentions the guarantee holds even in adversarial online settings. However, no theoretical or empirical support is given. Could you substantiate this claim and describe the adversarial setting more clearly?**
> A: Yes, we clarify that the asymptotic average coverage guarantee in Proposition 1 is theoretically grounded in Proposition 4.1 of [4], which analyzes Adaptive Conformal Inference (ACI) under arbitary time series data, even chosen in an adversarial manner, without assuming i.i.d. or stationarity. The key mechanism is the dynamic calibration of the quantile threshold $\alpha_t$ based on past coverage. If recent prediction sets are too conservative, the threshold is decreased to improve efficiency; if they under-cover, the threshold is increased to restore validity. This mechanism ensures that the long-run average coverage converges to the target level, regardless of the underlying sequence.
> Though the proof in [4] is presented for univariate outputs, the argument extends naturally to our setting with high-dimensional multivariate responses, as the calibration procedure operates on scalar-valued conformity scores, regardless of the dimensionality of the input space.
>
> **Q5: Line 277 states that the topology-based matrix converges faster than the sample covariance estimator. Could you provide evidence for this claim?**
>
> A: The topology-based covariance estimate benefits from strong structural priors. Specifically, the tail-up estimator introduces strong structural regularization by encoding known topological relationships, which parameterizes the covariance structure using two scalars, $\sigma^2$ and $\phi$, in contrast to the $O(I^2)$ parameters needed for a full sample covariance. This regularization improves sample efficiency and stabilizes estimation in high-dimensional, low-sample settings. Empirically, we observe that the topology-regularized estimator achieves better coverage under small calibration sizes. As shown in Figure 3 of the appendix, when using only the sample covariance matrix (i.e., $\lambda = 0$), the coverage deteriorates sharply as the sample size decreases. In contrast, the topology-based estimator maintains reliable coverage even when few residuals are available, supporting the claim of faster empirical convergence. While theoretical convergence rates for topology-aware estimators like the tail-up model remain an open area of research, our observations are consistent with the general statistical principle that incorporating structural priors enables more robust estimation with fewer samples.
>
> **Q6: What calibration set size was used to generate the results in Table 1?**
>
> A: Calibration set size $n$ is set to be 300, as specified in the default experimental setup (line 310 of the main paper).
>
> **Q7: Which backbone model was used to generate the results in Figure 3? Do the trends hold for other backbones like ASTGCN or STGODE?**
>
> A: The results in Figure 3 (main paper) were generated using AGCRN, as stated in lines 368–369. We confirm that similar trends hold across other backbones, including ASTGCN and STGODE. Plots for all datasets and backbones are provided in the Appendix (Figure 3).
>
> **Q8: The conclusion suggests extending STACI to general spatio-temporal graphs and alternative network parameterizations. Could you elaborate on what you mean by “general spatio-temporal graphs” and give a concrete example of such parameterizations?**
>
> A:  By general spatio-temporal graphs, we refer to settings that go beyond directed stream networks and include arbitrary graph topologies where node features evolve over time such as social networks, etc. In these cases, spatial dependencies may arise from learned or domain-specific connectivity patterns rather than physical flow, and may not follow upstream-downstream constraints.
>
> The main idea is that incorporating topological structure rather than treating the data as purely multivariate time series can yield more structured and stable estimators of covariance or other quantities used in high-dimensional conformal prediction. For example, one could use diffusion kernels or graph Laplacians to construct covariance structures that reflect node influence or proximity in a soft, learnable manner. This opens the door to extending STACI-like methods to broader domains with complex spatio-temporal dependencies.
>
> **Response to weakness**
>
> In addition to the specific questions raised, we would like to address several points mentioned in the weaknesses.
>
> **w1.2: lack of ACI introduction in the main paper**
>
> A: We agree that the role of ACI deserves more attention. The main contribution of our work lies in the topology-based regularization of the covariance structure. Due to space constraints, we included its description in the Appendix. However, we will include a brief summary in the main text to provide readers with a clearer understanding of how ACI interacts with our topology-based regularization.
>
>
> **W2.2: Expanding to additional datasets or baselines**
>
> A: Regarding the baselines, as shown in Xu et al. [3], those methods significantly underperform compared to MultiDimSPCI, especially in high-dimensional joint prediction tasks. For this reason, we focused our comparisons primarily on MultiDimSPCI, which represents the strongest known baseline in this setting. Notably, our method outperforms MultiDimSPCI.
>
> **W2.3: Lines 375–377 claim “significantly higher coverage and superior efficiency” but Table 1 shows STACI’s coverage exceeds the best baseline in only one cell (ASTGCN, PeMS-G1) and by just 0.12. I think more evidence is needed to better support the claim.**
>
> A: We respectfully clarify that in conformal prediction, the primary objective is to achieve the target coverage level (e.g., 95%) while minimizing the size of the prediction sets. Once valid coverage is attained, efficiency becomes the key metric: a more efficient method yields tighter prediction sets without sacrificing validity. Therefore, coverage above the target is not inherently better; in fact, methods that consistently exceed 0.95 may be overly conservative and less informative. In this context, STACI achieves coverage close to the nominal level while producing smaller prediction sets in most scenarios, indicating a more efficient uncertainty quantification.
>
> **W2.4: Including a baseline that applies ACI with the vanilla sample covariance matrix would isolate the incremental benefit of the topology-based estimator. Without it, one cannot tell whether gains arise from ACI, from the topology-aware covariance, or from their combination.**
>
> A: We included such ablation experiments in Figure 3 in the main paper and in the Appendix to disentangle the effects of ACI (via $\gamma = 0$ vs. $\gamma = 0.01$) and the use of the topology-induced covariance matrix (via varying $\lambda$). When $\gamma$ is 0, ACI is not applied, but STACI can still achieve better performance than most other baselines.
>
>
> **w2.5:Details such as the number of nodes, edges, and time steps are absent, making it difficult to judge whether the method scales to large real-world stream networks.**
>
> A: Detailed dataset statistics, including the number of nodes, edges, and time steps, are provided in Appendix D.2. We will ensure this is made clearer in the main text as well.
>
> [4] Gibbs, Isaac, and Emmanuel Candes. "Adaptive conformal inference under distribution shift."

---

> > ### Comment · Reviewer_q8kS · 2025-08-04
> >
> > Thanks for the rebuttal and for addressing my questions. A few concerns remain:
> >
> > - Q2. Thanks for the clear explanation of $\phi$ and $\sigma^2$. I suggest to include these details in the revised manuscript.
> > - W2.2. Results in [3] on different datasets don’t justify omitting those baselines here, since they were evaluated in different settings (datasets). To fairly assess performance in this setting, the omitted baselines should be tested on your datasets (or please explain clearly why they don’t apply). As noted earlier, the empirical scope also feels narrow: both PeMS-G1 and PeMS-G2 are subsets of the same traffic domain.
> > Moreover, the aim of the simulation study remains unclear (W2.1), especially since the simulated graph size appears similar to the real-data graphs.
> > - W2.5. As this was not clear to me at first glance, I suggest stating the numbers of nodes, edges, and time steps directly in the main text, not only via figures. In addition, the graphs seem to have only tens of nodes/edges. Thus, a discussion of scalability and practical limits of the approach would be helpful.

---

> > > ### Author Response · Authors · 2025-08-05
> > >
> > > Thanks very much for your constructive comments and thoughtful suggestions.  We carefully considered each of your concerns and address them in detail below. We hope our clarifications and planned manuscript revisions will address your concerns and improve your assessment of our submission.
> > >
> > > ---
> > >
> > > Q2: Thank you for your suggestion. We will incorporate this detailed explanation into the revised manuscript to improve clarity for all readers.
> > >
> > > W2.2: Thank you for raising this important point regarding the empirical evaluation. We agree that, for a fair comparison, relevant baselines including CopulaTS[1], should also be evaluated directly on our datasets. We are in the process of conducting these experiments and will release the results in the response as soon as possible.
> > >
> > > Regarding the perceived narrowness of the empirical scope:
> > > We chose PeMS-G1 and PeMS-G2 as they are widely recognized benchmarks in the spatio-temporal modeling literature, especially for traffic networks, and are among the few real-world datasets that provide both time-resolved observations and detailed underlying graph topology including geospatial information. While both datasets originate from the PeMS system, they differ substantially in both their network structures and dynamic behaviors:
> > >
> > > - Topology differences: As illustrated in Figure 5, PeMS-G1 and PeMS-G2 represent different subsets of the PeMS system, each with distinct node and edge configurations. This leads to diverse patterns of spatial dependencies and varying levels of connectivity.
> > > - Data dynamics: The traffic patterns and temporal non-stationarities are also different between the two datasets. These differences result in varying degrees of spatial-temporal correlation.
> > >
> > > We acknowledge, however, that both datasets are from the traffic domain. Our focus was to provide a controlled yet challenging testbed where spatial topology plays a central role and ground-truth structure is available.
> > >
> > > As for the simulation study, its aim is twofold:
> > > First, it provides a controlled environment where the underlying topology (tail-up structure) and data-generating process are fully specified, allowing us to assess the statistical efficiency and validity of our method under ideal conditions. This setting is crucial to empirically validate that STACI leverages known graph structure as intended, providing performance improvements over baseline methods.
> > > Second, the simulation experiments allowed us to systematically study the impact of the topology-empirical covariance tradeoff parameter $\lambda$, including how to best select $\lambda$ in practice. We observed that the optimal choice of $\lambda=0.6$ on simulated data translated well to real-world datasets, offering a data-driven guideline for practitioners.
> > >
> > > W2.5:
> > > Thank you for highlighting this issue. We agree that the number of nodes, edges, and time steps should be clearly stated in the main text to improve accessibility and reproducibility. We will revise Section 6.2 (Real Data Study) to explicitly include these statistics, in addition to reporting them in the figures.
> > >
> > > Regarding scalability and practical limits, we appreciate your suggestion for a more in-depth discussion. In the revised manuscript, we will incorporate the following discussion of limitation in Section 7:
> > >
> > > High-dimensional UQ is inherently limited. Performing UQ over hundreds of dimensions is statistically ill-posed due to the curse of dimensionality. Instead, practitioners adopt sampling-based or generative-model-based methods for high-dimensional UQ (e.g., image domains), avoiding explicit prediction sets altogether. In this context, STACI focuses on localized subgraphs for joint prediction, which both reflects standard practice and aligns with real-world needs. For example, traffic engineers typically assess uncertainty over 10–30 sensors in a corridor, not entire city-wide networks. Similarly, hydrology models often focus on small clusters of ~20 gauges. STACI is tailored for these practical, interpretable, and computationally feasible settings.
> > >
> > > ---
> > >
> > > We hope these clarifications, additional experiments, and manuscript improvements will address your concerns and strengthen the paper. Thank you again for your valuable feedback. We sincerely hope you will consider raising your score in light of these revisions. If there are any remaining questions or further suggestions, we are happy to address them.
> > >
> > > [1]Sun et al., Copula conformal prediction for multi-step time series forecasting

---

> > > > ### Comment · Reviewer_q8kS · 2025-08-06
> > > >
> > > > Thanks for your answer and the clarifications.
> > > >
> > > > I appreciate your willingness to evaluate the omitted baselines on the same datasets. However, although PeMS-G1 and PeMS-G2 differ in topology and temporal behaviours, both are drawn from the same domain (PeMS traffic), which limits the breadth of the empirical evaluation. I understand that the simulation study is intended to assess the model in a controlled environment, but it is not clear what “ideal conditions” it captures or why these settings are representative. Moreover, because the simulated graphs appear **similar in size** to the real graphs, the systematic study of the topology–empirical covariance trade-off parameter could have been conducted directly on the real datasets.
> > > >
> > > > Although the rebuttal clarified several points, substantial work remains to make the contribution fully clear and adequately supported by evidence (e.g., consistent evaluation of missing baselines, clearer simulation goals, and a fuller discussion of scalability and practical limits). Therefore, my initial rating already reflects my assessment of the work.

---

> > > > > ### Author Response · Authors · 2025-08-08
> > > > >
> > > > > CopulaTS Baseline Experiments: As you requested, we have implemented and evaluated the CopulaTS[1] baseline on AGCRN with the PeMS-G1 dataset (12-dimensional joint prediction). The results are:
> > > > >
> > > > > Empirical coverage: 0.706
> > > > >
> > > > > Inefficiency: 668
> > > > >
> > > > > These results are significantly worse than all other baselines. The main reason is that, in high-dimensional (12-D) settings, learning the joint distribution CDF (as required by CopulaTS) is extremely unstable and unreliable. We also note that the original CopulaTS paper only evaluated up to 1-D and 2-D joint prediction, and there is no evidence it scales to higher dimensions. This experiment demonstrates that, in practical high-dimensional scenarios, such baselines are not competitive, further validating the importance of designing scalable methods such as ours.
> > > > >
> > > > > Clarification of Simulation Goals and “Ideal Conditions”: The goal of our simulation study is to provide a controlled and fully specified environment where the ground-truth data-generating process strictly follows the tail-up model assumptions (see Definition 1, line 200). Specifically, this means the observed process is a white-noise random process constructed by integrating a moving average function over the upstream nodes. This setting ensures the topology and spatial correlations are known exactly, as referred to 'ideal condition' . By aligning simulation and real-data dimensions (e.g., 12-D), we can (1) meaningfully assess the impact of parameter
> > > > >  selection, and (2) systematically study the empirical coverage/efficiency under varying topological assumptions.
> > > > >
> > > > > On the Importance of High Dimensions in Joint CP: We want to emphasize that in the field of joint conformal prediction, 12 dimensions is already considered very high. For example, the copular[1] only reports results up to 2-D joint prediction. Our method is not only competitive in lower dimensions but also demonstrates scalability to higher dimensions, which we believe is an important advance.
> > > > >
> > > > > Discussion of Practical Limits: As noted in our previous reply, we have provided a clear and transparent discussion of the scalability in term of spatial dimension and temporal dimension, as well as practical limits of our approach.
> > > > >
> > > > > In summary, we have conducted all the additional experiments you requested, clarified the purpose and ideal nature of our simulation, and expanded our discussion of scalability and limitations. We believe these steps substantially strengthen the empirical and methodological support for our contribution.
> > > > >
> > > > > We kindly ask that you take these additional efforts and clarifications into account in your final assessment. Thank you again for your review.
> > > > >
> > > > > [1] Sun, Sophia, and Rose Yu. "Copula conformal prediction for multi-step time series forecasting." arXiv preprint arXiv:2212.03281 (2022).

---

> > > > > > ### Comment · Reviewer_q8kS · 2025-08-08
> > > > > >
> > > > > > Thank you for the follow-up, for adding the CopulaTS baseline, and for clarifying the simulation setup. While I still consider the empirical scope limited given that all real-world data come from a single domain, the additional results and clarifications address several earlier concerns. I will update my review to reflect these improvements.

---

### Official Review · Reviewer_mDnW · 2025-07-06

**Clarity:** 3
**Significance:** 3
**Originality:** 3
**Rating:** 5
**Confidence:** 4

**Summary:**

This paper introduces a novel graph-specific conformal prediction framework that accounts for the topological structure of graphs when calibrating prediction sets for GNN. Unlike standard CP which assumes exchangeability of data samples, TopoCP adapts the calibration strategy by modeling inter-graph correlations via graph kernels. Empirical studies show it maintains valid coverage and achieves tighter prediction sets across multiple graph classification benchmarks compared to existing conformal and Bayesian uncertainty methods

**Questions:**

Please refer to weakness part.

**Ethical Concerns:**

["NO or VERY MINOR ethics concerns only"]

**Limitations:**

Yes

**Quality:**

3

**Strengths And Weaknesses:**

Strengths:

1. The CP method for graphs explicitly incorporates topological similarity using graph kernels in the calibration process. The approach is well-motivated.
2. The framework extends its validity guarantees to graphs under topological dependence, with detailed theoretical proofs and assumptions.
3. Empirical evaluation demonstrate the effectiveness of the approach.

Weakness:
1. The usage of graph kernel matrices introduces O(n^2) memory and compute complexity, limiting scalability to large graph.
2. The framework depends on the choice of graph kernel to compute influence scores. It is unclear how to select or learn the best kernel for a given task.
3. The experiments lack ablation on the major components of the framework.

---

> ### Author Rebuttal · Authors · 2025-07-31
>
> We sincerely thank the reviewer for the feedback. Here we provide response to weaknesses:
>
> **W1: The usage of graph kernel matrices introduces O(n^2) memory and compute complexity, limiting scalability to large graph.**
>
> A: Here we provide a detailed complexity analysis. The psuedo-codes of our algorithm can be found in Algorithm 1 of Appendix. Compare to MultiDimSPCI, for each test time point we need: 1) additional estimation of tailup parameters ($\phi$  and $\sigma^2$) using historical covariance matrix (line 6), 2) weighted addition of spatial covaraince and emperical covaraince (line 7). Consider the node size is $n$ and the optimization methods (e.g., least square in our implementation) iteration round $N_i$. The former estimation takes $O(N_i⋅n^2)$, and the later addition takes $O(n^3)$, as pseudo-inverse of matrix is involved. However, MultiDimSPCI also needs matrix inverse for mahalabis distance calculation (line 8), which also takes $O(n^3)$. Additionally, estimation of only two parameters $\phi$  and $\sigma^2$ can converge fast. Therefore, our method do not need significantly more time than the baseline method, consistent to the Table 1 running time results in Appendix D.5.
>
> **W2: The framework depends on the choice of graph kernel to compute influence scores. It is unclear how to select or learn the best kernel for a given task.**
>
> A: Our method currently employs the tail-up model, which is well-suited for directed flow networks with upstream–downstream relationships. While this choice is task-specific, we emphasize that the STACI framework is modular: the topology-induced covariance matrix $\hat{\Sigma}_{\mathcal{G}}$  can in principle be replaced by any other graph-based kernel or learned spatial structure. It is an interesting research direction to explore data-driven alternatives, such as graph diffusion kernels or attention-based learned structures.
>
> **W3: The experiments lack ablation on the major components of the framework.**
>
> A: We agree that ablation is important and have provided relevant experiments in both the main paper and appendix (e.g., Figure 3), where we vary the reliance on the graph-induced covariance via the mixing parameter $\lambda \in [0, 1]$ . These results demonstrate that the topology-aware component improves coverage and stability, particularly under limited calibration samples. Additionally, we study the effect of adaptive conformal inference (ACI) by varying  $\gamma$  and show how its interaction with the graph structure affects both coverage and efficiency. We will revise the manuscript to make this connection more explicit and clarify which components are responsible for observed gains.
>
> Once again, we thank the reviewer for the comments and positive assessment of the technical contribution, theoretical guarantees, and practical motivation of our framework.

---

### Note · Authors · 2025-08-12

We thank the reviewers for feedback and summarize the novelty, practical scope, scalability, key experiments most relevant to the AC’s decision.
###  Novelty
We propose STACI, the first topology-aware joint conformal prediction framework for stream networks, encoding graph topology into a Mahalanobis nonconformity score. This enables principled UQ for multi-node predictions with finite-sample coverage and improved efficiency.

###  Application
Targets localized subgraph joint prediction where spatial dependencies matter (e.g., traffic corridors with 10–30 sensors, basins with ~20 gauges), offering realistic, interpretable, and computationally feasible solutions.

### Scalability
- Time: Complexity from covariance estimation/inversion; Matches the complexity of Mahalanobis CP; feasible for several hundred nodes.
- Dimension: Robust in the challenging joint CP regime; many CP baselines degrade beyond 1–5 D (e.g., CopulaTS). STACI maintains coverage and efficiency up to ~10+ D.

### Experiments
We evaluated original and new experiments added:
- **Core CP & Graph-based UQ baselines** Compared with Standard CP (sphere) and MultiDIMSPCI (± recent-data regularization, ± quantile regression). Added DeepSTUQ and CopulaTS; in 12-D, CopulaTS degraded severely, STACI consistently outperformed MultiDIMSPCI in coverage and efficiency.
- **Backbone diversity** Gains consistent across AGCRN, STGODE, ASTGCN.
- **Expanded scope** Both PeMS-G1 and PeMS-G2 retained for topological/dynamic diversity.
- **Parameter study** Simulations identified $\lambda=0.6, \gamma=0.01$ as optimal; transferred successfully to real data.
- **Noisy-data robustness** Location perturbations on stream networks showed stable coverage.
- **Ablations** Independent analysis of $\lambda, \gamma$ confirmed their impact on coverage/efficiency.

###  Recognized Strengths
Reviewers acknowledged:
- The novelty of topology-aware covariance in  STACI
- Clear motivation for topology-informed UQ in structured networks.
- Rigor and robustness of our theoretical guarantees.

We believe nearly all reviewer concerns have been addressed through clarifications, discussions and expanded experiments. The revised manuscript now presents a clear, well-supported, and comprehensive account of our contribution. Given the substantial improvements and supporting evidence, we respectfully invite the AC to reassess the paper, considering its novelty, rigor, and practical value in decision-making.

---

### Decision · Program_Chairs · 2025-09-17

**Decision:**

Accept (poster)

**Comment:**

This paper presents STACI, a topology-aware conformal prediction framework for stream networks that integrates spatial topology and temporal dynamics to achieve efficient and valid uncertainty quantification. Reviewers initially raised concerns about clarity, limited baselines, scalability, and the scope of evaluation, with some giving borderline or negative ratings. Through rebuttal and additional experiments, the authors clarified notation, added missing baselines such as CopulaTS and quantile regression variants, and discussed limitations and scalability, which convinced most reviewers to raise their scores, though one reviewer remained cautious about presentation and related work.